# Turn air-captured CO$_2$ with methanol into amino acid and pyruvate in an ATP/NAD(P)H-free chemoenzymatic system

Jianming Liu [1,3], Han Zhang[1,3], Yingying Xu[2], Hao Meng [2] & An-Ping Zeng [1] ✉

The use of gaseous and air-captured CO$_2$ for technical biosynthesis is highly desired, but elusive so far due to several obstacles including high energy (ATP, NADPH) demand, low thermodynamic driving force and limited biosynthesis rate. Here, we present an ATP and NAD(P)H-free chemoenzymatic system for amino acid and pyruvate biosynthesis by coupling methanol with CO$_2$. It relies on a re-engineered glycine cleavage system with the NAD(P)H-dependent L protein replaced by biocompatible chemical reduction of protein H with dithiothreitol. The latter provides a higher thermodynamic driving force, determines the reaction direction, and avoids protein polymerization of the rate-limiting enzyme carboxylase. Engineering of H protein to effectively release the lipoamide arm from a protected state further enhanced the system performance, achieving the synthesis of glycine, serine and pyruvate at g/L level from methanol and air-captured CO$_2$. This work opens up the door for biosynthesis of amino acids and derived products from air.

Photosynthesis as a form of biosynthesis using CO$_2$ from atmosphere and solar energy has shaped the birth and past of life on the earth. Technical biosynthesis using gaseous and air-derived CO$_2$ will certainly shape the future of our planet and human being in view of the ever-increasing anthropogenic emission of CO$_2$, which threatens to break the balance of planetary climate[1,2]. For the sustainable development of human beings, we are facing further global challenges, such as the supply of goods and food for an ever-growing population, which calls for using CO$_2$ as an abundant carbon source with the input of sustainable energy to replace fossil carbon and even carbohydrates for manufacturing of commodities[3].

Among various proposed solutions, biosynthesis using CO$_2$ has received great attention owing to its mild conditions, broad product spectrum and high selectivity compared to chemical use of CO$_2$[4–6], which normally demands expensive catalysts and suffers from narrow product variety[7]. But the drawbacks of biosynthesis include low driving force and limited CO$_2$ fixation rate. An appealing solution is to combine biological and chemical (non-enzymatic) syntheses and harness their respective advantages. A representative example is to integrate

the use of energy-rich one-carbon compounds (C1) as carbon and energy source, such as methanol and formate that can be produced from CO$_2$ efficiently using renewable energy, into microbial metabolism[8–13]. Among the different natural and artificial CO$_2$ fixation pathways, the reductive glycine pathway (rGlyP) has received particular attention, because it is considered as one of the most efficient C1 (formate) assimilation pathways[14]. The core part of rGlyP is the reversible glycine cleavage system (rGCS) consisting of four enzymes: T protein (aminomethyltransferase), P protein (glycine decarboxylase), L protein (dihydrolipoamide dehydrogenase) and the aminomethyl carrier H protein. The rGlyP has been implemented into different microorganisms and their performance is still much limited due to the low C1 metabolic flux, slow microbial growth and poor biomass yield[8–10].

One approach to overcome the limitations of microbes utilizing CO$_2$ is to achieve CO$_2$ fixation in cell-free synthesis, as demonstrated for starch biosynthesis from methanol[15], C2 (Acetyl-CoA or glyoxylate) production from bicarbonate[16,17]. While this approach displayed impressive results in terms of CO$_2$ (or equivalents) fixation rates and

[1]Center of Synthetic Biology and Integrated Bioengineering, School of Engineering, Westlake University, 600 Dunyu Road, Xihu District, Hangzhou 310024 Zhejiang Province, China. [2]Beijing Advanced Innovation Center for Soft Matter Science and Engineering, Beijing University of Chemical Technology, Beijing, China. [3]These authors contributed equally: Jianming Liu, Han Zhang. ✉e-mail: zenganping@westlake.edu.cn

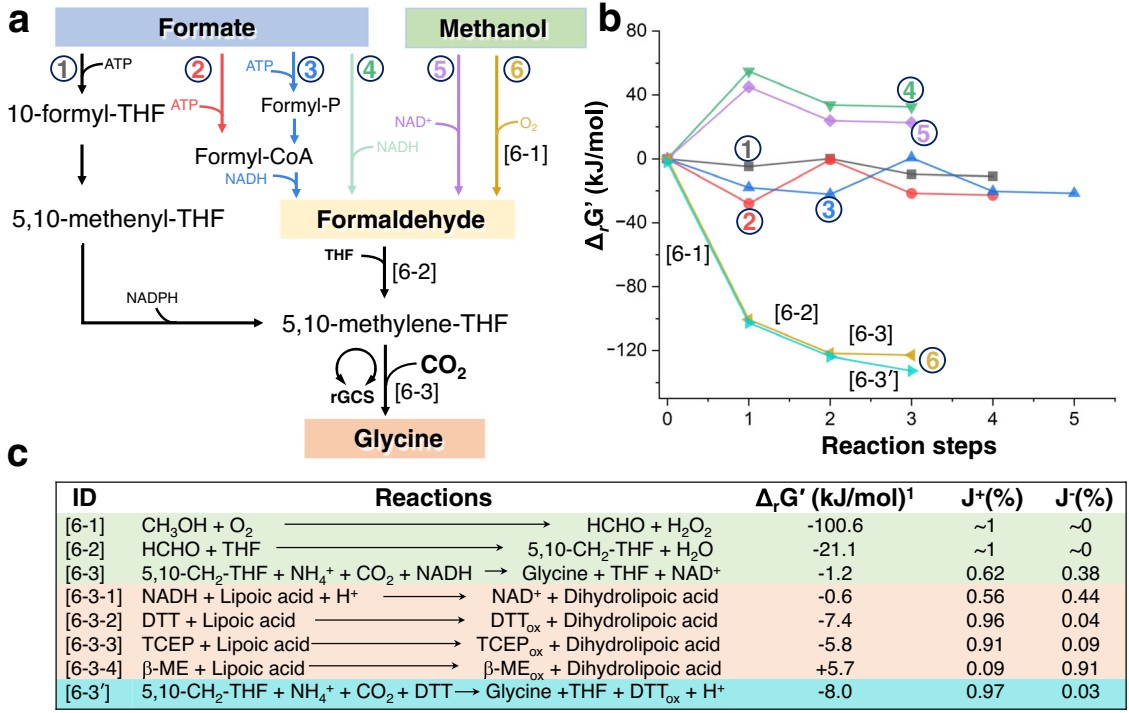

**Fig. 1 | Thermodynamic characterization of different glycine biosynthetic pathways and corresponding reactions. a** The six metabolic pathways leading to the synthesis of glycine. **b** Thermodynamic profiles of the Gibbs free energy change for individual steps of the pathways under physiological conditions (pH 7.5, ionic strength I = 0.20, substrates = 1 mM). **c** Thermodynamic and kinetic features of individual reaction steps in the sixth pathway from methanol to glycine. [6-1], methanol oxidation. [6-2], formaldehyde (HCHO) and THF condensation. [6-3], glycine synthesis. [6-3-x], lipoic acid reduction by different reducing agents. DTT Dithiothreitol, TCEP Tris(2-carboxylethyl)phosphine, β-ME β-mercaptoethanol, THF tetrahydrofolate, Formyl-P formyl phosphate, rGCS reverse glycine cleavage system. [1]$\Delta_r G'$=-RTln(J$^+$/J$^-$), J$^-$, the forward flux in glycine synthesis, J$^-$, the backward flux in glycine cleavage, the fraction in the enzymatic flux is indicated.

diverse metabolic flexibilities, they have some common drawbacks, including the requirement of expensive energy ATP and reducing powers NAD(P)H, many reaction steps (normally more than ten)[18], the occurrence of thermodynamically less favorable reactions, and thus severely limited overall reaction rate.

Here, we report a designed ATP and NAD(P)H-independent catalytic system that efficiently converts $CO_2$ and methanol into the C2 amino acid glycine. It is based on an engineered rGCS with the NAD(P)H-dependent L protein replaced by biocompatible electron-shuttle chemistry with dithiothreitol. DTT-based chemistry is able to enhance the thermodynamic driving force by breaking the bio-thermodynamic boundary with an increased $\Delta_r G'$ from −1.2 kJ/mol to −8.0 kJ/mol, avoid thus the need for NADH and increase the carboxylase activity, finally leading to more than 50-fold enhancement in glycine production rate. Engineering of H protein to effectively release the lipoamide arm from a protected state further enhanced the system performance, achieving the synthesis of glycine, serine and pyruvate at g/L level from methanol and air-derived $CO_2$. This work demonstrates the impact of integrating chemical and biological catalysis and protein engineering in developing a chemoenzymatic system for the biosynthesis of amino acids and derived products from air.

## Results

### Chemical break of the biological thermodynamic limitation
We set out to first synthesize glycine (C2) from gaseous $CO_2$ with a simultaneous assimilation of formate or methanol, which can be synthesized from $CO_2$ abundantly[19], as a liquid C1 carbon and renewable energy source to achieve that both two carbon atoms in glycine can be derived from $CO_2$ molecules. We evaluated the thermodynamics of six different pathway combinations by calculating the change of Gibbs

free energy in each reaction of the individual pathway (Fig. 1, Supplementary Tables 3, 4). The $\Delta_r G'$ of each step in rGlyP (Pathway 1) approaches zero under physiological conditions, underlying that it is highly reversible and thus not favorable for effective glycine synthesis. This also applies to Pathways 2 and 3, which use formaldehyde as an intermediate derived from formate and have a similar thermodynamic profile. Formaldehyde as an energy-rich active C1 synthon is principally of high interest for biosynthesis. We have thus further considered three possible formaldehyde-relevant pathways (Pathways 4–6). Pathways 4 and 5 are not feasible due to the large barriers for formate reduction and methanol dehydrogenation to generate formaldehyde. On the other side, the large negative $\Delta_r G'$ value in pathway 6, especially for the first reaction step, indicates a strong driving force in transforming methanol into formaldehyde, which pushes the whole reaction into the glycine synthesis direction.

Pathway 6 consists of methanol oxidation with alcohol oxidase, formaldehyde condensation with tetrahydrofolate (THF) and the rGCS. Thermodynamic analysis of rGCS obtained from eQuilibrator or MetaCyc[20,21], reveals a $\Delta_r G'$ value of −1.2 kJ/mol, implying that it is highly reversible and almost 40% of the reaction flux is in the reverse (glycine cleavage) direction (Fig. 1c)[22]. To debottleneck this thermodynamic limitation, we examined whether chemistry could be used to provide a higher driving force and to abolish the need of biological cofactors. We paid special attention to the reduction of lipoic acid via opening of a disulfide bond that is the only reaction step in rGCS involving NADH with a $\Delta_r G'$ value of −0.6 kJ/mol.

In chemistry, the reduction of a typical disulfide bond can be performed by using reducing agents, such as dithiothreitol (DTT, Fig. 2b), Tris(2-carboxylethyl)phosphine (TCEP, Fig. 2c), and β-mercaptoethanol (β-ME). We calculated $\Delta_r G'$ of lipoic acid reduction

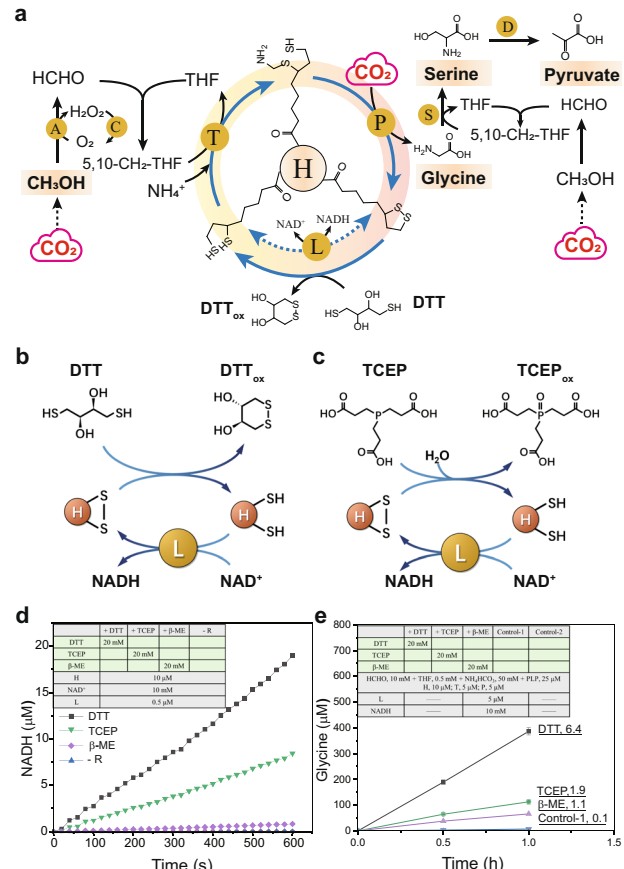

**Fig. 2 | Design and implementation of an ATP and NAD(P)H-free CO₂-fixation pathway via biocatalysis and biocompatible non-enzymatic chemistry. a** The reaction system to convert $CO_2$ and methanol into C2 glycine, C3 serine and pyruvate. DTT is used to replace the function of L protein (dotted line). **b, c** DTT and TCEP as potential reducing agents to reduce $H_{ox}$ to $H_{red}$ as a non-enzymatic reaction step. **d** Measurement of NADH generation coupled with the disulfide-reducing chemistry. The dynamic change of $H_{ox}$ to $H_{red}$ in the reaction catalyzed by reducing agents (R) is measured using NADH generation in the presence of L protein and NAD⁺. **e** Glycine production rate underlined (mM/min) with the components listed. Data represent mean values ± SEM of 3 independent experiments. 5,10-CH₂-THF, $N_5$, $N_{10}$-methylene-THF; A, alcohol oxidase, EC 1.1.3.13; $H_{int}$, aminomethylated form of the lipoylated H protein; $H_{ox}$ and $H_{red}$, oxidized and reduced form of the lipoylated H protein, respectively; PLP, pyridoxal 5-phosphate monohydrate; P, glycine decarboxylase, EC 1.4.4.2; T, aminomethyltransferase, EC 2.1.2.10; L, dihydrolipoyl dehydrogenase, EC 1.8.1.4; S, serine hydroxymethyltransferase, EC 2.1.2.1; D, L-serine deaminase, EC 4.3.1.17.

using these three reducing agents. DTT exhibits the highest driving force with a $\Delta_r G'$ of −7.4 kJ/mol, indicating that 96% of the flux can be in the forward reaction (glycine synthesis). Taking the whole rGCS together, the replacement of L-protein and NADH by using DTT drops the $\Delta_r G'$ value to −8.0 kJ/mol, suggesting that suitable chemistry is able to break the thermodynamic biological boundary and determine the reaction direction (Fig. 1c).

## Design and implementation of an ATP and NAD(P)H-free CO₂ fixation pathway

Based on the favorable thermodynamic profile of Pathway 6 (Fig. 1b), we decided to implement this **i**ntegrated **c**hemo**e**nzymatic **C**O₂ to **a**mino-acid **p**athway (ICE-CAP) (Fig. 2a). We experimentally determined the reducibility of $H_{ox}$ to $H_{red}$ using chemical reducing agents. To monitor the dynamic change of $H_{ox}$ and $H_{red}$ (oxidized and reduced form of the lipoylated H protein) in the reaction process, we coupled the non-enzymatic chemical reduction of $H_{ox}$ to $H_{red}$ with NADH

generation by L protein. It was found that DTT exhibited the best NADH formation rate, followed by TCEP (Fig. 2d), and the use of sole β-ME did not show detectable NADH formation, all of which were in accordance with the thermodynamic analysis. We also measured the consumption of DTT directly using Ellman's reagent and it unanimously agreed with the electron transfer from DTT to $H_{ox}$ (Supplementary Fig. 1).

After that, we assembled the rGCS system by replacing L protein and NADH with DTT-dependent non-enzymatic chemistry. The glycine formation rate was 0.1 μM/min in the presence of L protein and NADH, but increased to 6.4 μM/min when DTT was introduced to replace L protein and NADH (Fig. 2e), followed by TCEP (1.9 μM/min), demonstrating that DTT is a fast-reducing and suitable agent for accessible disulfide bridges under the experimental conditions. In addition, we quantified the conversion rate from $H_{ox}$ to $H_{red}$ with DTT. Most of the $H_{ox}$ (50 μM) was converted to $H_{red}$ within 5 min using 1 mM DTT with a conversion rate of roughly 10 mmol $H_{red}$/min, which is much larger than 0.67 mmol $H_{red}$/min using L. The increase of L concentration from μM to mM level did not enhance the production rate of glycine (Supplementary Figs. 2–4). This approach to replace enzymatic reaction of L using biocompatible non-enzymatic chemistry allows the continuous reduction of $H_{ox}$ without the use of expensive NADH and also creates a favorable condition for glycine biosynthesis from both thermodynamic and kinetic perspectives.

## Enhanced glycine synthesis through the change of redox state of P protein

Due to the excellent property of DTT in reducing disulfide bonds, it is frequently used to prevent the aggregation of cysteine-containing proteins. This motivated us to investigate if DTT could additionally stimulate enzymatic activities and enhance glycine synthesis in the rGCS system.

In rGCS, P protein is responsible for the crucial carboxylation step[23]. The crystal structure of P protein from *Synechocystis* sp. PCC6803, which has a high amino acid similarity (57%) (Supplementary Fig. 5) to its *E. coli* counterpart, shows that the C terminus is locked in a closed conformation by a disulfide bond between C972 in the C terminus and C353 in the active site. According to literature[23], the disulfide bridge can prevent the binding of ligands in the active site and therefore reduce the enzymatic activity of P protein. Based on the predicted protein structure of P protein from *E. coli* K12 (Uniprot: P33195) using AlphaFold2[24], we pinpointed the corresponding cysteine locations C335 in the active site and C949 (or C951) in the C terminus, and modeled the oxidized (close, $P_{ox}$) and reduced state (open, $P_{red}$) (Fig. 3a). In order to observe the two states of P protein, a direct quantification method was developed for the analysis of P protein, and it clearly displayed two peaks in the absence of DTT. When DTT was added, the fraction of the reduced state increased (Fig. 3b).

To test the potential roles of cysteines in forming the disulfide bond and their effects on glycine synthesis, we created P protein mutants C335S, C949S, C951S and C949S/C951S and assayed them in the absence and presence of DTT, respectively. In the absence of DTT, the variant C335S showed a more than sixfold decrease in glycine formation rate, because the C335 residue is located in the active site of P protein and its mutation led to a dramatic decrease in glycine synthesis, which is in consistent with the observations achieved using C353 mutants of cyanobacterial P protein[23]. However, the C949S and C951S mutants led to an approximately threefold increase in glycine formation rate (Fig. 3c). The activity of C949S/C951S with double mutations did not differ from that of C949S. Therefore, the disruption of the disulfide bridge through mutation of C949S or C951S contributed to the increased glycine synthesis. In the presence of DTT, wild type P protein ($P_{wt}$) exhibited more than 50-fold improvement in glycine synthesis compared to the condition without DTT, but the introduced single or double mutations in P protein resulted in

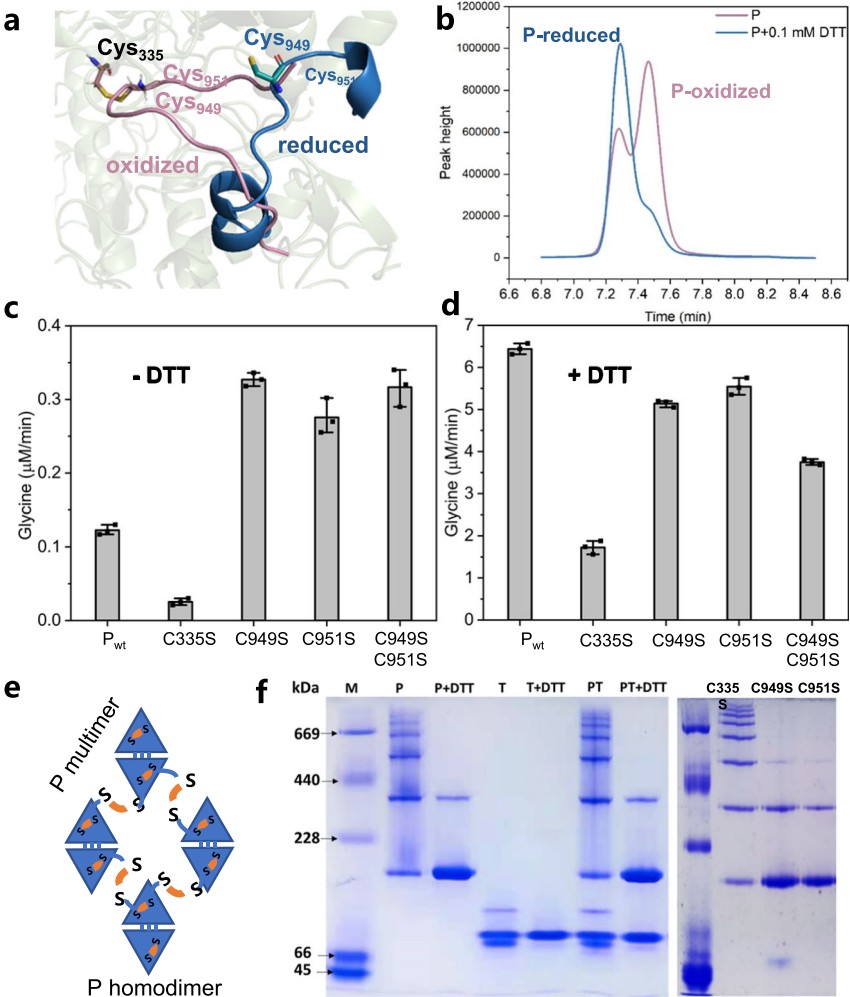

**Fig. 3 | Enhanced glycine synthesis through the change of redox state of P protein. a** The oxidized state of P protein with the formation of disulfide bond between C335 and C949 (or its neighboring C951) and its reduced state, when the disulfide bond was open. **b** Quantification of P protein using HPLC. **c** Glycine production rate using P protein or different variants in the absence of DTT. **d** in the presence of DTT. Bars represent the mean change ± SEM of 3 independent experiments. **e** The disulfide bond-crosslinked network in P proteins. **f** Native PAGE analysis of P, T protein with (without) DTT and P variants (One representative figure is shown in 3 independent experiments).

decreased glycine formation (Fig. 3d). The latter is contrary to the observations found in the absence of DTT. This indicates that DTT, in addition to regenerating $H_{red}$ from $H_{ox}$, also plays a key role in regulating the redox state of P protein and this observation fits well with the stimulation of cyanobacterial P activities by DTT[23]. They presented that the disulfide bridge between C353 (equivalent to C335 in *E. coli* P protein) and C972 (equivalent to C949 or C951 in *E. coli* P protein) in cyanobacterial P protein was reduced with DTT and the free C terminus was folded away from the active site entrance, leading to enhanced P protein activities[23]. To confirm the enhancement of P activities by DTT specifically in our reaction system, we measured the conversion of $H_{ox}$ to $H_{int}$ catalyzed by P protein and it was found that this rate was increased by adding DTT (Supplementary Fig. 6). In cyanobacterial P protein, when the disulfide bond between C353 and C972 was reduced, the pyridoxal phosphate (PLP) cofactor could get into the active center of P protein followed by its conformational changes, such as C terminus extension that might interact with H protein. Similarly, the mutation of C949S or (and) C951S located at the C terminus of *E. coli* P protein might influence its mobility and this in turn could influence substrate binding and lead to poor catalysis (Fig. 3d). The change in conformation also alters the surface charge through electrostatic surface analysis, which becomes remarkably more positive (Supplementary Fig. 7). The closed loop conformation allows the basic

residues, Arg315, Arg321, Arg326, and Arg327, to be fully exposed to the surrounding environment. They form a positively charged surface close to the active site entrance, which might be a beneficial driving force to absorb negatively charged bicarbonate.

In addition to the observed intramolecular disulfide formation, we speculated whether P protein and other proteins could form polymerization due to the formation of disulfide bond-crosslinked networks (Fig. 3e). Through native PAGE analysis, we indeed observed the obvious formation of P multimers and this network can be largely destroyed by DTT (Fig. 3f). The introduction of C335S led to more pronounced P multimers, while C949S and C951S resulted in less polymerization. This indicates that the C terminal cysteine residues C949 and C951 play important roles in the formation of disulfide networks. Apart from P protein, T protein also assembles into T multimers and the polymerization is significantly changed in the presence of DTT (Fig. 3f, Supplementary Figs. 8–10). Therefore, the formation of disulfide bonds largely occurs inside P and T proteins and the addition of DTT can change their redox states and enhance the production of glycine.

### Increased glycine synthesis by breaking the self-protection of H protein lipoamide

The lipoamide arm in H protein in the forms of $H_{ox}$ and $H_{red}$ is normally located on the surface of H protein[25]. But in $H_{int}$, it is locked in the

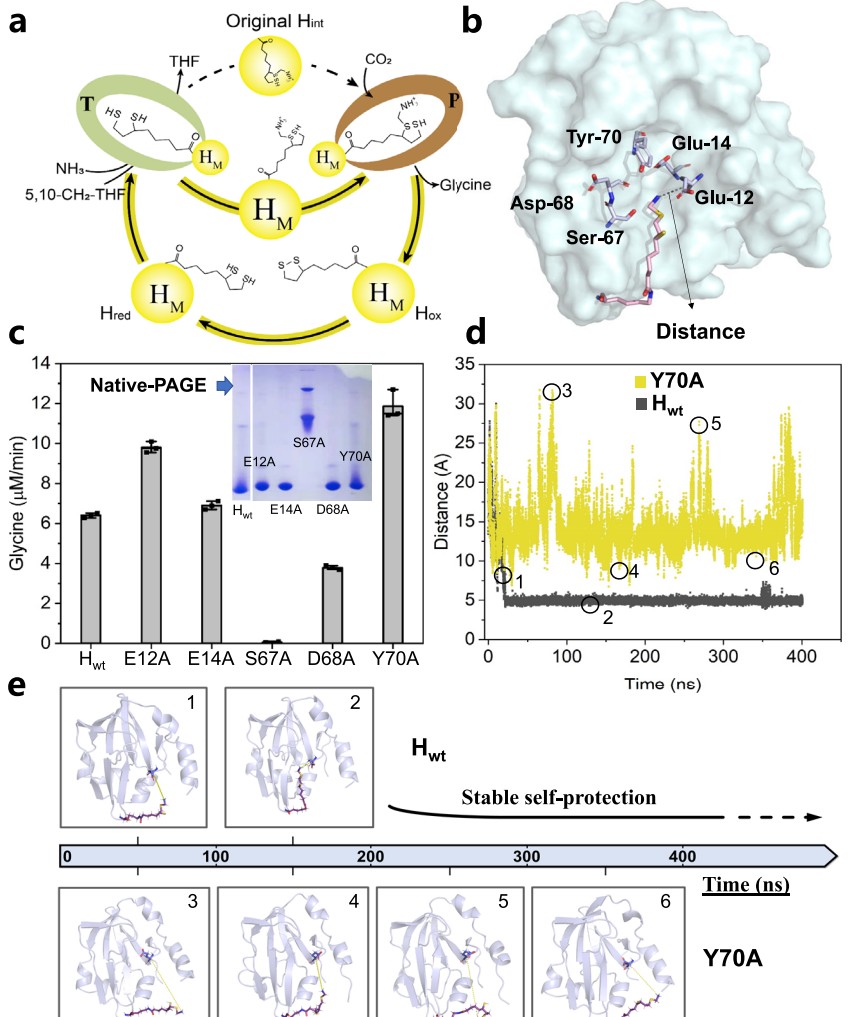

**Fig. 4 | Self-protection of H protein and increased glycine synthesis by breaking it. a** The lipoamide arm is engaged as a mobile substrate that commutes successively between the other two enzymes and breaking its self-protection in the original $H_{int}$ can bypass the process in which the swinging arm shuttles inside and outside of the H cavity. $H_M$, H mutants. **b** Key amino acids important to maintain the aminomethyl arm in the H cavity. **c** The glycine production rate using different H variants (10 mM) and their corresponding native PAGE analysis. Bars represent the mean change ± SEM of 3 independent experiments. **d** Molecular dynamics simulations for $H_{wt}$ and Y70A. The distance between Ca at E12 of the H protein and the N atom of the lipoamide, shown in Fig. 4b, is measured for the first 400 ns in the simulations. **e** The representative structures in MD trajectory for $H_{wt}$ and Y70A as labeled in Fig. 4d.

bottom of H cavity through electrostatic interactions and hydrogen bonds[25, 26]. This self-protection of the aminomethyl arm inside the cavity can prevent the non-enzymatic release of $NH_3$ and toxic formaldehyde due to nucleophilic attack by water molecules[27], which is an evolutionary advantage for living organisms. While this process is critical for intracellular metabolism, it is undesired in the cell-free biocatalysis process as the self-protection of lipoamide arm can increase the overall reaction time. It is reported that the spontaneous degradation of aminomethyl moiety was relatively slow[28].

The lipoamide arm is engaged as a mobile substrate that commutes successively between T and P (Fig. 4a). To destroy its self-protection in $H_{int}$, we selected several key amino acids that are crucial to maintaining its docked conformation in the H cavity[29] and mutated them all to alanine, individually (Fig. 4b). We found that the variants of E12A and Y70A, enhanced their glycine formation rates significantly, whereas S67A and D68A showed the opposite effect (Fig. 4c). Native PAGE analysis showed that S67A displayed a different protein structure, which might be caused by protein aggregation leading to an almost complete loss of its activity, while the other variants are similar to the wild type. We then performed molecular dynamics (MD)

simulations for $H_{wt}$ and Y70A to examine the trajectory of lipoamide arm swinging into the cavity and its release. The distance between α-carbon at E12 of the H protein and the N atom located in the aminomethyl moiety was measured for the first 400 ns in simulations. For $H_{wt}$, the distance was concentrated at around 5 Å after the initial fluctuations for roughly 20 ns, which means that the aminomethyl lipoate arm is locked into a highly stable conformation in $H_{wt}$. However, this distance is much broader for Y70A, ranging from 5 to 35 Å (Fig. 4d), indicating that the stable location of the lipoamide arm in the cavity was destroyed to a large extent according to the MD trajectory analysis (Fig. 4e). Since the mutant at Y70 position performed the best in glycine production rate, we then created a Y70 saturated mutagenesis library. All the Y70 variants, except for Y70C, showed better activities than $H_{wt}$. Generally, when the polar amino acid Y is substituted by nonpolar amino acids A, V, F, W, P or positively charged amino acids R, K (the aminomethyl arm is positively charged), the glycine production rate increased significantly, especially for Y70W, Y70P, Y70R, and Y70F with approximately more than two folds (Supplementary Fig. 11). We also performed MD simulations for these four mutants and they all demonstrated that the self-protection of the aminomethyl arm was

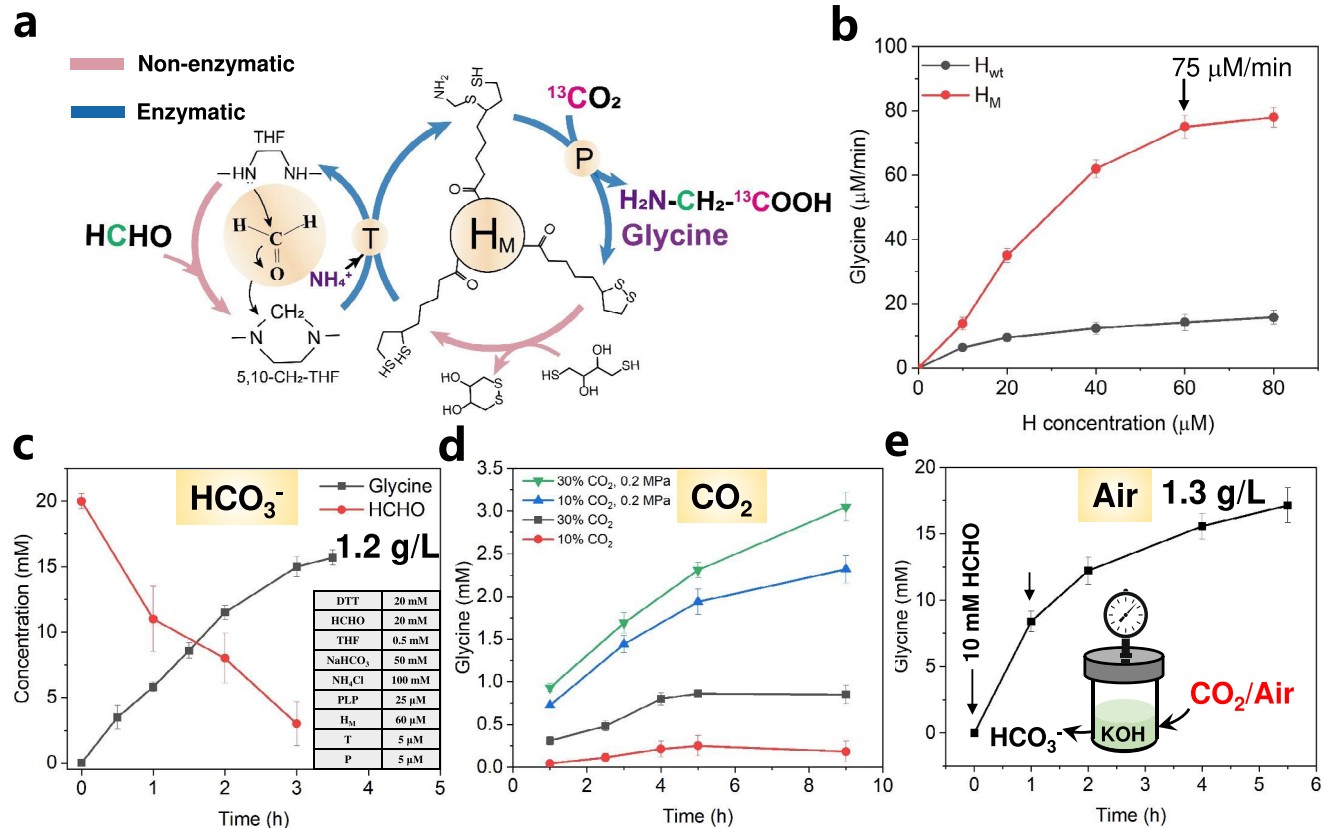

**Fig. 5 | Glycine synthesis from formaldehyde and CO₂. a** The combination of non-enzymatic and enzymatic chemistry to synthesize glycine. **b** The effect of H concentrations on glycine production rate. **c** Glycine production from formaldehyde and bicarbonate. **d** Glycine production from formaldehyde and gaseous $CO_2$. **e** Glycine production from formaldehyde and air-derived $CO_2$. 10 mM formaldehyde was added in the beginning followed by another 10 mM formaldehyde after 1 h to avoid high levels of precipitation caused by the reaction between a high concentration of formaldehyde and ammonium. All the Data represent mean values ± SEM of 2 independent experiments.

destroyed or altered to different extents, individually (Supplementary Fig. 12). As the negative control, Y70Q displayed a similar trajectory and catalytic activity as $H_{wt}$. But in Y70C, Y70M and D68A, although the lipoamide arm was escaped from the hydrophobic cavity, they were trapped by the bottom α-helix of H protein in another location shown in supplementary Fig. 13 and this trap has not been observed for all the other mutants in Y70W, Y70P, Y70R, and Y70F.

**Conversion of gaseous or air-derived CO₂ with formaldehyde to glycine**

Based on the DTT-dependent chemistry and H mutants ($H_M$ refers to Y70P, Y70R or Y70W, respectively, Y70R is used in the following experiments), we assembled a system consisting of two non-enzymatic and two enzymatic reactions to first convert bicarbonate (C1) and formaldehyde (C1) into glycine (C2) (Fig. 5a). The condensation of formaldehyde with THF is a spontaneous non-enzymatic reaction[30] and its product 5,10-CH₂-THF is responsible for C1 supply and THF regeneration in rGCS. We optimized the concentration of H protein and found that it had a significant influence on glycine production rate that reached as high as 75 μM/min at 60 μM $H_M$, but it was only less than 20 μM/min for $H_{wt}$ (Fig. 5b). We subsequently optimized the formaldehyde concentration and in a concentration range from 10 to 20 mM, it resulted in a high glycine production rate above 70 μM/min (Supplementary Fig. 14), which can be further enhanced to -100 μM/min by optimizing the ratio of C/N. We noticed that when the concentration of formaldehyde was more than 20 mM, precipitation was formed similar to the observation in previous studies[31]. This precipitation was caused by the interaction of high levels of formaldehyde and ammonium[31,32]. Furthermore, we tested

different concentrations of DTT on the effect of glycine production under the optimal condition. It was found that 20 mM DTT supplemented in solution (40 mM DTT in total, as 20 mM DTT comes from THF solution) performed the best (Supplementary Fig. 15). We also measured DTT stability in the reaction system and <2% DTT was oxidized in 8 h (Supplementary Fig. 16), which demonstrates its robustness in potential scale-up applications.

To confirm glycine synthesis from CO₂ and formaldehyde, $NaH^{13}CO_3$ or (and) $H^{13}CHO$ labeling approach was applied to determine carbon assimilation. The $m/z$ = 218 and $m/z$ = 246 fragments in glycine derivative clearly confirmed that the carbon source of glycine (−COOH, −CH₂) was derived from CO₂ and formaldehyde, respectively, with carbon labeling levels in glycine approaching around 100% under each condition (Supplementary Fig. 17, 18). The mole ratio of glycine to CO₂ is 1:1. Using the combination of formaldehyde and bicarbonate, we obtained 15.5 mM (1.2 g/L) glycine in 3.5 h with a high carbon yield of 78% based on formaldehyde and 31% based on CO₂ (Fig. 5c).

Direct use of gaseous CO₂ is appealing[33]. We tested different concentrations of gaseous CO₂ ($v/v$): 30% CO₂ with 70% air, 10% CO₂ with 90% air as direct carbon input for the engineered rGCS system. At atmospheric pressure, we achieved 0.8 mM and 0.2 mM glycine, respectively. To increase the concentration of dissolved gas in the solution, we increased the external pressures and found that glycine production was significantly enhanced to 3.0 mM and 2.3 mM for 30% and 10% CO₂ under 0.2 MPa, respectively (Fig. 5d). Further increase of the pressure to 0.5 MPa did not lead to a higher production (Supplementary Fig. 19). We also fed air containing around 420 ppm CO₂ in this system for 10 h but no glycine was detected. Because the CO₂ concentration in air is extremely low, the development of carbon

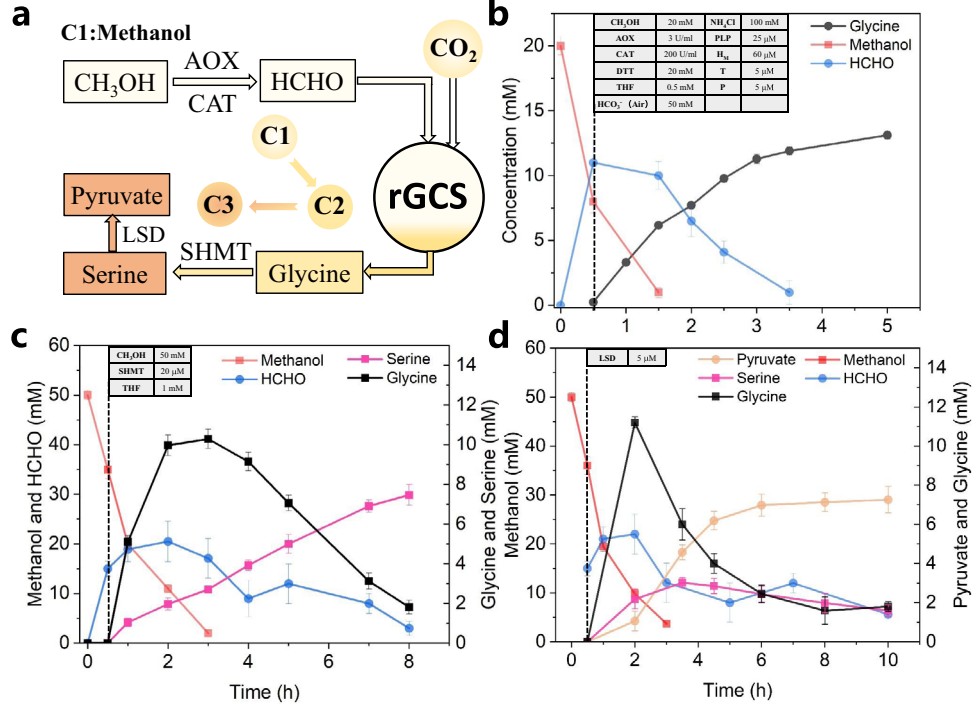

**Fig. 6 | Turn methanol and CO₂ into C2 (glycine) and C3 (serine and pyruvate).** **a** The process of converting C1 to C2 and C3. **b** Methanol and air-captured CO₂ as carbon source for glycine production with the components listed in the embedded table. **c** The biosynthesis of serine using the same components as listed in (**b**) with 50 mM methanol, 1 mM THF and 20 mM SHMT. **d** The production of pyruvate using the same components in (**c**) with 5 mM LSD. All the Data represent mean values ± SEM of 2 independent experiments.

capture and utilization (CCU) process is essential and valuable, as only the direct removal of $CO_2$ from air can actually reduce the global atmospheric $CO_2$ concentration. We then tested the combination of chemical absorption with biological synthesis for developing an integrated CCU process. We investigated direct air capture by bubbling air (420 ppm $CO_2$) through a 20 mL solution of 1 M KOH. After 72 h, 4 mmol $CO_2$ was captured, resulting in a pH of around 8.6 equal to ~200 mM bicarbonate. Then it was diluted into the rGCS system at 50 mM as carbon substrate with additional formaldehyde (10 mM formaldehyde was supplemented at first followed by another 10 mM formaldehyde after 1 h to avoid precipitation). Finally, 17.1 mM (1.3 g/L) glycine was achieved in 5.5 h (Fig. 5e) with a high production yield of 86% based on formaldehyde and 34% on $CO_2$.

### Conversion of methanol and air-derived CO₂ into glycine and serine and pyruvate

In ICE-CAP (Fig. 2a), methanol is selected as a liquid C1 carbon and energy source to capture $CO_2$. Different enzymes can convert methanol into formaldehyde: alcohol dehydrogenase and alcohol oxidase (AOX). Since the conversion by AOX is an efficient thermodynamically favorable process (Fig. 1) and no $NAD^+$ is required[15], we selected AOX from *Pichia pastoris* and optimized its concentration. Using 3 U AOX and 200 U CAT, 9.8 mM formaldehyde was formed in 0.5 h (Supplementary Fig. 20). Then the rGCS system with air-captured bicarbonate was added to initiate glycine production that finally reached 13.2 mM (1.0 g/L) in 5 h (Fig. 6b) with the production yield at 66% based on methanol and 26% on $CO_2$.

Glycine as a C2 platform chemical can be readily extended to C3 chemicals, for example, serine and pyruvate. The conversion of glycine to serine requires serine hydroxymethyltransferase (SHMT, EC 2.1.2.1) and another C1 donor 5,10-CH₂-THF (Fig. 2a). Using ¹³C-labeling experiments, we confirmed that this carbon donated by 5,10-CH₂-THF is added into the β-carbon position of serine (Supplementary Fig. 21). We first optimized the concentration of SHMT from *E. coli* and

20 μM SHMT gave rise to 5.2 mM serine from 10 mM glycine in 2 h (Supplementary Fig. 22). Then the rGCS and 20 μM SHMT were supplemented after 0.5 h reaction between methanol and AOX. It was observed that glycine reached 10.3 mM in 3 h, followed by a gradual decrease to 1.8 mM. The final serine concentration arrived at 7.5 mM (0.8 g/L) in 8 h (Fig. 6c) with the production yield at 30% of theoretical maximum from methanol and 15% from $CO_2$. Since we added 50 mM methanol in the reaction system, which was able to generate more than 20 mM formaldehyde during the reaction process, this high concentration of formaldehyde led to the formation of precipitate and therefore caused a lower production yield.

Serine can be further extended into the platform chemical pyruvate through harnessing L-serine dehydratase (LSD, EC 4.3.1.17). We compared Fe-S dependent LpLSD from *Legionella pneumophila* and PLP-dependent MsLSD from *Mycolicibacterium smegmatis*[34]. MsLSD was more robust and turned 10 mM serine to pyruvate in an almost complete version in 1 h. The fresh-made LpLSD enzyme showed similar activities, but lost half of its activity after 1 week storage (Supplementary Fig. 23), which might be caused by the sensitivity of Fe-S cluster to oxygen[35]. So based on the serine production system, 5 μM MsLSD were introduced together with the rGCS, including SHMT, and 6.9 mM (0.6 g/L) pyruvate were produced in 5 h (Fig. 6d) with a comparable production yield to serine. We noticed that pyruvate production was plateaued at around 4 h and this might be caused by the toxicity of formaldehyde on MsLSD (Supplementary Fig. 24). It was found that MsLSD was quite sensitive to formaldehyde: 10 mM formaldehyde caused a loss of half of the MsLSD activity, while 20 mM formaldehyde caused a 75% activity loss. SHMT is more robust and it can tolerate a high concentration of formaldehyde (the condensation of formaldehyde and THF is the substrate for SHMT). At 50 mM formaldehyde, the activity of SHMT maintained at 75% of its activity compared to that achieved at 10 mM formaldehyde. Increasing the tolerance of enzymes towards formaldehyde and controlling its concentrations through process engineering can further enhance the production of these C3 compounds.

## Discussion

This study demonstrates the development of an ATP and NAD(P)H-free chemoenzymatic system for amino acid biosynthesis from gaseous and air-captured $CO_2$. The energy and electrons in ICE-CAP used to fix $CO_2$ come from C1 donor and DTT. This double C1 fixation strategy, which uses an electron-intensive C1 (methanol) to capture electron-deficient C1 ($CO_2$), offers an efficient manner to take full use of the input energy. DTT-based chemistry plays an essential role in linking the metabolic pathway between P and T enzymatic reactions to substitute NADH-required L protein. DTT also creates a highly favorable condition for glycine synthesis through debottlenecking the biological thermo-dynamic limitation, which makes the reversible rGCS almost non-reversible and proceeds predominantly in the glycine formation direction. Furthermore, DTT can prevent P protein polymerization and enhance glycine production. For scale-up, DTT regeneration in a sustainable manner is desirable for cell-free synthesis. It has been demonstrated that DTT can be recycled electrochemically and $DTT_{ox}$ is able to get electrons to regenerate DTT from a tungsten cathode in neutral aqueous solutions at a potential of −1.0 V (vs. saturated calomel electrode). This regenerated DTT in turn can serve as an excellent electron mediator and reduce lipoamide by thiol disulfide interchange[36,37].

In addition to DTT-dependent chemistry, another non-enzymatic step lies in the spontaneous reaction between formaldehyde and THF with a high in vitro reaction rate[30], channeling methanol through AOX into the rGCS system. We obtained a glycine production rate of 53.7 nmol/min/mg from methanol and $CO_2$ (Table 1). This rate is the highest $CO_2$-fixation rate reported so far, which is 10-fold higher than that of the synthetic crotonyl–coenzyme A (CoA)/ethylmalonyl-CoA/hydro-xybutyryl-CoA (CETCH) cycle and 2.5-fold higher than that of the artificial starch anabolic pathway (ASAP). Therefore, by harnessing the synergy of chemistry and biocatalysis, it expands the solution in creating efficient approaches for developing CCU processes.

Direct air capture is an engineered process to directly reduce a considerable amount of accumulated $CO_2$ from the atmosphere. In recent years, it has undergone significant technical development with commercial entities operating in the market[38]. However, its large-scale application suffers from techno-economic viabilities. The capture of $CO_2$ into bicarbonate and its following conversion to amino acids offer a solution for $CO_2$ valorization, while it is still very much in its infancy. Further development of ammonium hydroxide as a sorbent can generate ammonium bicarbonate that is a perfect carbon and nitrogen source for amino acid biosynthesis, which will further reduce the purification cost of sorbents and benefit the process feasibility.

Through ICE-CAP, we achieved the transformation of abundant C1 ($CO_2$, methanol, formaldehyde) to C2 (glycine) and C3 (serine and pyruvate). In addition to serving as valuable products, these C2 and C3 compounds can also be excellent feedstocks or building blocks to produce more diverse long-chain carbon commodities[39]. As traditional substrate for biosynthesis is based on sugars, which raises concerns about food security and biodiversity. Therefore, exploring the use of renewable non-food resources is becoming important for industrial biomanufacturing and the utilization of these feedstocks paves the way toward a circular carbon economy.

## Methods

### Materials

Glycine, nicotinamide adenine dinucleotide ($NAD^+$, NADH), Tris (2-carboxyethyl) phosphine (TCEP), THF and AOX enzyme (Pichia pastoris, A2404) were obtained from Sigma-Aldrich (St. Louis, MO, USA). The derivatization reagents dansyl chloride and CAT enzyme ($10^5$ U/ml) were purchased from Yuanye Bio-Technology Co., Ltd. (Shanghai, China). Dithiothreitol (DTT), β-mercaptoethanol (β-ME), pyridoxal 5'-phosphate monohydrate (PLP) and flavin adenine dinucleotide (FAD) were obtained from Aladdin (Shanghai, China). *Escherichia coli* DH5α and BL21 (DE3) were used for plasmid construction and the

**Table 1 | Comparison of cell-free C1 ($CO_2$) fixation pathways**

| Pathway | Main Substrate | Main Product | ATP/CO₂ (mol/mol) | NAD(P)H/CO₂ (mol/mol) | Number of enzymes | O₂ | C1 fixation rate (nmol/min/mg) |
|---|---|---|---|---|---|---|---|
| Formolase[47] | More than 40 mM Formate | 0.4 mM Dihydroxyacetone Phosphate | 1 | 1 | 5 | Robust | – |
| CETCH[17] | 0.2 mM Propionyl CoA + 50 mM NaHCO₃ | 0.54 mM Glyoxylate | 1 | 2 | 17 | Robust | 5.0 |
| POAP[48] | 20 mM Acetate Sodium + 100 mM NaHCO₃ | 0.023 mM Oxalate | 1 | 0.5 | 4 | Sensitive | 8.0 |
| ASAP[15] | 100 mM Methanol | 1280 mg/L Starch | 0.5 | 0 | 11 | Robust | 22.0 |
| rGPS-MCG[16] | 0.4 mM Crotonyl CoA + 0.4 mM PEP+ 100 mM NaHCO₃ | 0.4 mM Glycerate + 1 mM Malate | 2.5 | 3 | 19 | Robust | 28.5 |
| ICE-CAP | 20 mM Formaldehyde + 50 mM HCO₃⁻ (air) | 17.1 mM Glycine (5.5 h) | 0 | 0 | 3 | Robust | 67.8 |
| | 20 mM Methanol + 50 mM HCO₃⁻ (air) | 13.2 mM Glycine (5 h) | 0 | 0 | 5 | Robust | 53.7 |

overexpression of recombinant proteins, respectively. $Ni^{2+}$-NTA resin was purchased from Genscript (Nanjing, China). Amicon® Ultra-15 filtration devices (molecular size cut-off 10 KDa for H protein, 30 KDa for T protein and L protein and 100 KDa for P protein) were purchased from Millipore (Billerica, MA, USA). Mut Express II Fast Mutagenesis kit V2 was purchased from Vazyme (Jiangsu, China). BCA protein assay kit and sodium dodecyl sulfate-polyacrylamide gel electrophoresis (SDS-PAGE) gels were purchased from SolarBio Co., Ltd. (Beijing, China). Luria-Bertani (LB) medium containing tryptone (10 g/L), yeast extract (5 g/L) and NaCl (10 g/L) were used for cloning and expression.

### Enzyme preparation
The plasmids and bacterial strains used in this experiment are given in Supplementary Table 1. Oligonucleotide sequences of primers used for cloning target proteins are shown in Supplementary Table 2. The genes coding for H, P, T, and L proteins were amplified from *E. coli* K12 genomic DNA, then cloned into the expression vector pET28a (NdeI and XhoI). The gene encoding SHMT was amplified from *E. coli* K12 genomic DNA and ligated with pET22b using NdeI and HindIII. The genes coding for LpLSD and MsLSD were codon-optimized for *E. coli* and synthesized by Vazyme (Supplementary Table 5). *E. coli* BL21 (DE3) harboring the resulting constructs was cultured in LB medium supplemented with 50 mg/L of kanamycin or 100 mg/L of ampicillin at 37 °C until the $OD_{600}$ of the culture reached 0.6-0.8. Isopropyl-beta-D-thiogalactopyranoside (IPTG) was added to a final concentration of 0.2 mM to induce protein expression for 12 h at 30 °C. The plasmid pET28a-P or pET28a-H was used as a template to generate mutations using Mut Express II Fast Mutagenesis kit V2, respectively. Lipoylation of H-protein was performed during its over-expression in vivo. To this end, lipoic acid (200 μM, pH 7.0) was added to the culture of the strain containing the plasmid pET28a-H or a H protein variant, prior to starting the cultivation to directly obtain lipoylated H-protein ($H_{lip}$). Following the over-expression, enzymes were purified as described previously[34]. The purified enzymes were checked by SDS-PAGE (Supplementary Fig. 3).

### Electron transfer reaction between $H_{ox}$ and $H_{red}$ with the presence of L-protein
The interconversion of $H_{ox}$ and $H_{red}$ was performed according to an enzymatic assay using an excess amount of a reductant for the reduction of the H protein bound lipoic acid, and the produced $H_{red}$ was then re-oxidized by L-protein in the presence of $NAD^+$. For the assay, the reaction mixture contained Tris-HCl buffer (50 mM, pH 7.5), 20 mM TCEP or 20 mM DTT or 20 mM β-ME, 10 μM $H_{ox}$ and 0.1 μM L-protein. The assay was initiated by the addition of 10 mM $NAD^+$. The rate of NADH formation was determined spectrophotometrically at 340 nm.

### Measurement of glycine production rate
The reaction mixture containing Tris-HCl (50 mM, pH 7.5), 20 mM DTT, 0.5 mM THF, 10 mM formaldehyde, 25 μM PLP, 5 μM P-protein, 5 μM T-protein, and 10 μM $H_{ox}$, named the rGCS solution, was used in most cases unless otherwise mentioned. The reaction was initiated by adding 50 mM $NH_4HCO_3$ to the reaction mixture and carried out at 37 °C for 1 h. Samples were taken periodically to measure the glycine formation using HPLC and calculate the glycine production rate. To evaluate the effect of different concentrations of formaldehyde, 1–30 mM formaldehyde were tested in the rGCS system. For optimization of the C/N ratio, 50 mM $NaHCO_3$ with different levels of $NH_4Cl$ were used together to replace $NH_4HCO_3$. Regarding glycine production from methanol, 20 mM methanol was reacted with AOX and CAT for 0.5 h and then the rGCS solution (without formaldehyde, detained components listed in the figure) was added to initiate the glycine synthesis process.

### Glycine production from gaseous $CO_2$
We tested different concentrations of gaseous $CO_2$ (v/v): 30% $CO_2$ with 70% air, 10% $CO_2$ with 90% air, and 100% air containing 420 ppm $CO_2$.

The rGCS solution without bicarbonate was mixed in a parallel 50 ml pressure reactor (WATTCAS, China) to a final volume of 10 ml. The reaction was started by the introduction of gaseous $CO_2$ in the presence of 0.0, 0.2, and 0.5 MPa, respectively. The solution was recovered from the reactor at different time points and analyzed using HPLC. Control reactions with air led to no detectable amounts of glycine.

### Serine and pyruvate production
For serine and pyruvate production, we increased the initial concentration of methanol to 50 mM and THF to 1 mM. 50 mM methanol was reacted with AOX and CAT for 0.5 h and then the rGCS solution (without formaldehyde, detained components listed in the corresponding figure) with 20 μM SHMT was added to initiate the serine synthesis process. Similarly, 2 μM LSD was added together with the rGCS solution and SHMT for pyruvate synthesis.

### Analytical methods using HPLC
The rGCS proteins, including H, P and T proteins were analyzed based on the HPLC method developed in our group previously and refined in this work[40]. The analysis was performed using a Shimadzu LC-2030C system with a ZORBAX 300SB-C18 column (4.6 × 250 mm, 5 μM) and monitored at 210 nm using a DAD-3000 diode array detector (Dionex, Sunnyvale, USA). The mobile phase consisted of acetonitrile (A) and 0.1% trifluoroacetic acid aqueous solution (B). The volume percentage of buffer B was varied as follows: linearly increased from 45% to 58% (0–5 min) and held at 58% for 5 min, then sharply increased from 58% to 90% (13.4–13.41 min), held at 90% for 3 min, and then sharply decreased to 45%, held at 45% to 23 min. The flow rate was 1.0 mL min$^{-1}$. BCA quantification kit was used to quantify the concentration of proteins and to establish the calibration curves for using HPLC to measure these proteins.

Glycine and serine concentration in the reaction mixture were determined by pre-column dansyl chloride derivatization. To this end, 40 μL of a reaction mixture was mixed with 160 μL of 0.2 M $NaHCO_3$ and 200 μL of 5.4 mg mL$^{-1}$ dansyl chloride in acetonitrile. Derivatization occurred at 30 °C for 30 min. After the reaction, 600 μL of 0.12 M HCl was added to adjust the pH of the sample to become weak acidic. After centrifuging at 10,000 × g the supernatant was filtered with 0.22 μm membrane. The dansyl derivative of glycine was measured using HPLC on a Shim-pack GIST $C_{18}$ column (5 μm, 4.6 × 150 mm) at 30 °C, with a mobile phase composed of acetonitrile and 20 mM potassium phosphate buffer pH 6.0 (25:75 v/v) at a flow rate of 0.8 mL/min. The effluent was monitored at 254 nm using a DAD. To double check the amino acid concentration, an Agilent InfinityLab Poroshell HPH-C18 4.6 × 100 mm, 2.7-Micron column (Agilent, USA) and Agilent 1260 Infinity HPLC system at DAD 338 nm after OPA derivatization (Agilent, USA) were used to measure the concentration of glycine and serine. The detailed separation conditions were as described by Agilent (Amino Acid Analysis: https://www.agilent.com/cs/library/applications/5994-0033EN-us-agilent.pdf).

Methanol was quantified by gas chromatography as reported previously[12] and pyruvate was quantified on HPLC with a Aminex HPX-87H column (Bio-Rad, Hercules, USA) and a DAD detector. The column oven temperature was set at 60 °C and the mobile phase consisted of 5 mM $H_2SO_4$, at a flow rate of 0.5 ml/min[41].

### Determination of $^{13}C$-labeled glycine, serine and pyruvate
To confirm carbon source of glycine, we carried out $NaH^{13}CO_3$, $H^{13}CHO$, and $NaH^{13}CO_3 + H^{13}CHO$ labeling experiments, respectively. The carbon donor in the reaction system containing Tris-HCl (50 mM, pH 7.5), 20 mM DTT, 0.5 mM THF, 10 mM HCHO, 50 mM $NaHCO_3$, 100 mM $NH_4Cl$, 25 μM PLP, 5 μM P-protein, 5 μM T-protein, and 60 μM $H_{ox}$, was replaced by the corresponding 10 mM $H^{13}CHO$, 50 mM $NaH^{13}CO_3$ and their combinations. Samples were taken after 1 h

reaction and were treated for GC-MS analysis according to the protocol of Long et al.[42].

The GC-MS analysis was carried out on an GC–MS (Agilent, No. 8890B) equipped with an Agilent HP-5MS capillary column (30 m, 0.25-mm i.d., 0.25 μm), connected to a mass spectrometer (Agilent, model no. 7077B). The injection volume was 1 μL with helium at 1 mL/min as the carrier gas and samples were injected in split mode at 25:1. The oven temperature program was set as follows: 80 °C for 2 min, followed by a ramp of 10 °C min$^{-1}$ to 280 °C, 10 min hold. The injection, transfer line, and ion source temperatures were 230, 280, and 250 °C, respectively. The accurate mass spectrometry data were acquired in full-scan mode ($m/z$ 50–600) after a solvent delay of 9 min. The MassHunter Workstation Version 10.0 software with NIST2020 library was used for raw peaks exacting, peak alignment, deconvolution analysis, peak identification and integration of the peak area. Quantification of labeling enrichment was done in SIM (single ion monitoring) mode using glycine fragment ($m/z$) of 218 and 246, serine fragment ($m/z$) of 288, 302, and 390.

### Quantification of formaldehyde

Formaldehyde was quantified by the acetylacetone method[15]. A 250 μL sample was reacted with 25 μL acetylacetone solution (0.02 M acetylacetone, 0.05 M acetic acid, 2 M ammonium acetate). After incubation at 60 °C for 15 min, the mixture was cooled down, centrifuged and analyzed at 414 nm.

### Standard procedure of $CO_2$ capture from air

The synthetic air contains a mixture of 80% ($v/v$) $N_2$ with 20% ($v/v$) $O_2$ and 420 ppm $CO_2$, stored in an aluminum alloy gas cylinder under 9.5 MPa. It was bubbled through a 1 M KOH (20 mL) solution in a parallel pressure reactor. The amount of $CO_2$ captured was calculated through gravimetric analysis of the solutions before and after the capture.

### Native PAGE

The rGCS proteins, including H, T, P, L proteins and their variants or mixtures were mixed with 5×Native-PAGE protein loading buffer (Real-Times Biotechnology Co., Ltd., Beijing, PL111) and loaded into wells of 6% or 15% (only for H protein) separating gels configured for low molecular weight proteins (Real-Times Biotechnology Co., Ltd., Beijing, RTD6130). Samples were electrophoresed by soaking in 1×Tris-Gly buffer at 150 V for 60 min. Following electrophoresis, the gel was stained with Coomassie blue 250 and decolorized with 30% methanol and 10% acetic acid. The concentrations of each protein in Native PAGE gel: M, Protein marker; P, 4.3 μM P protein; P + DTT, 4.3 μM P protein + 50 mM DTT; T, 10.8 μM T protein; T + DTT, 10.8 μM T protein + 50 mM DTT; PT, 4.3 μM P protein + 10.8 μM T protein; PT + DTT, 4.3 μM P protein + 10.8 μM T protein + 50 mM DTT; C335S, 4.3 μM; C949S, 4.3 μM; C951S, 4.3 μM. The original gel figures were provided in Supplementary Fig. 25.

### Molecular dynamics simulations

We performed the molecular dynamics simulations using the Amber20 package[43] with the Amber14 force field. The parameters including charges (AM1-bcc) and atom types for the lipoamide were generated using Antechamber provided in Ambertools21. PyMOL was used for visualization. For each simulation, the system with 1.0 nm of padding was filled with TIP3P water molecules and neutralized with $Na^+$ counter ions. Each system was minimized using the steepest descent of 5000 steps and conjugate gradient methods of 5000 steps to eliminate any overlap or clash between the atoms. Convergence was reached when the maximum force was not >1000 kJ mol$^{-1}$ nm$^{-1}$ at any atom. Then the system was heated up to 300 K using the Langevin dynamics. At this stage, position restraints on the heavy atoms of the protein allowed the surrounding water molecules to relax. When the system temperature was stabilized, the system was equilibrated at a constant pressure of 1 bar using the Berendsen Barostat method. A simulation was run for 2 ns until the density was stable. After that, simulations were performed using an integration time step set to 2.0 fs. All hydrogen-bonds were constrained using the SHAKE algorithm[44]. For each system, 15 independent trajectories for 1 ns starting from random number seeds were simulated. Afterward, clustering analysis was used to choose a "representative" structure. The "representative" structure is the conformation with the smallest average root-mean-square deviation from the centroid of the largest population. Subsequently, we simulated the "representative" structure for the following simulations to obtain the trajectory.

### Electrostatic potential surface calculation for $P_{ox}$ and $P_{red}$

The structure of $P_{ox}$ was modeled using SWISS-MODEL workspace[45]. The structure of $P_{red}$ was based on the AlphaFold2 protein structure of P protein from *E. coli* K12 (Uniprot: P33195). The electrostatic potential surface was calculated using APBS[46] and visualized in PyMOL.

### Preparation of THF solution

The stock solution of THF is 10 mM. To prepare 1 ml solution, a suitable amount of THF powder is first dissolved in 30 μL 1 M KOH solution. After it is completely solubilized, 270 μL $H_2O$ and 300 μl 0.2 M $KH_2PO_4$ are added in the solution, subsequently. Finally 400 μL 1 M DTT is added to prevent its oxidation.

### Reporting summary

Further information on research design is available in the Nature Portfolio Reporting Summary linked to this article.

## Data availability

The data that support the findings of this study are presented in Supplementary Information. And the source data underlying Figs. 1b, 2d, e, 3b–d, 4c, d, 5b–e, 6b–d, and Supplementary Figs. 1b, 2, 4, 11, 14, 15, 16, 18, 19, 20, 22, 23, 24 are provided in Source Date file with this paper or available from the corresponding author on request. Source data are provided with this paper.

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

## Acknowledgements

This work was financially supported by the National Key R&D Program of China (2022YFA0912003), the Competitive Research Funding Program (WU2022A004) in Center for Synthetic Biology and Integrated Bioengineering at Westlake University and Westlake Education Foundation. We thank Dr. Runqing Yang, Borong Zhang, Quanchi Lv, Xiaoze Ying, Aocong Guan and Yudian Zhu for being involved in the establishment of analytic methods for GCS proteins and for assisting some of the experiments. We also thank Prof. Ping Wang for helpful suggestions on the thermodynamic analysis. We thank the High-Performance Computing Center at Westlake University for the software resources and computing resources.

## Author contributions

L.J. designed and performed the experiments, analyzed data, and wrote the manuscript. Z.H. worked in the MD simulations and the rGCS protein analysis. Y.X. and H.M. worked in the in vitro experiments. Z.A.-P. conceived, designed, and supervised the project and wrote the manuscript.

## Competing interests

The authors declare no competing interests.
