## [Peer Review File · Nature Communications]

Turn air-captured CO₂ with methanol into amino acid and pyruvate in an ATP/NAD(P)H-free chemoenzymatic systemREVIEWER COMMENTS

Reviewer #1 (Remarks to the Author):

Based on the reductive glycine pathway, the author demonstrated a cell-free, ATP and NADH-independent CO₂ fixing system to produce glycine and serine from captured CO₂ (bicarbonate) and methanol. To attain NADH-independent CO₂ fixation, they replaced NADH with DTT to provide a stronger reducing power for glycine cleavage system (GCS). The results showed that DTT can improve the reduction rate of the H protein and simultaneously elevate carboxylase activity of the P protein through reducing disulfide bonds. Nevertheless, the kinetic activity of the wild-type H protein is not high enough for efficient glycine production. As a result, they engineered the lipoamide arm of H protein to generate a Y70A variant with a 2-fold increase in activity. Implementing the engineered protein, the DTT driven enzymatic system can convert bicarbonate and methanol into glycine at a rate of 77.8 nmol/min/mg, which is the highest CO₂ fixation rate to date among reported cell-free systems.

Major comments:

1. Although the authors claimed their cell-free system achieved the highest CO₂ fixation rate reported to date, the novelty and scientific insight of this work is limited. Fundamentally, the research team demonstrated an enzymatic reaction of reductive glycine pathway that can be driven by DTT in vitro. Using a chemical reducing agent to serve as an electron source is not sustainable. The problem cofactor regeneration is now shifted to DTT.
2. Efficient cofactor regeneration in a sustainable manner is critical for a cell-free system to fix CO₂ at scale. Can the authors discuss possible potential ways to transfer electron from sustainable source to DTT? Also is DTT stable enough to be used in the cell-free system aerobically?
3. The title of this manuscript and the claims at Line 109 are misleading. Their system cannot directly use CO₂ as sole feedstock for amino acids production.
4. The use of term "air-derived CO₂" may be imprecise. More importantly, emphasizing on the direct use of "air-derived CO₂" in this article is not appropriate. They performed chemical absorption to capture CO₂ instead of direct air capture by enzymes, and the chemical reaction unit was totally separated from the enzymatic module.

Minor comments:

1. Line 102: The statement of "The latter is able to enhance the thermodynamic driving force by breaking the bio-thermodynamic boundary....." is unclear.
2. Fig. 6a: The diagram in the middle of figure is confusing. C₁ chemicals cannot be converted to C₃ chemicals directly in their work.
3. The authors can conduct ¹³C-labelling experiments to support that all the amino acids were derived from C₁ feedstock.

Reviewer #2 (Remarks to the Author):

The manuscript describes the in vitro application of the reversible glycine cleavage system (rGCS) for the conversion of the C1 compounds CO₂ and methanol into the C2 compound glycine (as well as C3 compounds serine and pyruvate). Remarkably, the central pathway consists of only 4 core enzymes and achieves the conversion of abundant methanol, CO₂ and ammonia into an amino acid at the expense of dithiothreitol (DTT) with high conversions. The study compares the rGCS pathway to other CO₂ converting in vitro pathways. The study thoroughly investigates the effect of DTT on the reduction of the lipoic acid as well as its secondary regulatory effect on the enzymes involved. Overall, the study shows in the beginning, how an in vitro pathway with the alcohol oxidase and the DTT dependent rGCS pathway shows favorable thermodynamics in comparison to natural occurring in vivo options. It then pursues to investigate and implement the in vitro pathway with satisfying results. Bottlenecks are removed very methodically by optimizing reaction conditions and apply enzyme engineering. It represents an elegant example in the emerging field of in vitro biosynthesis and underlines the capability of chemoenzymatic processes.

Specific comments:

Fig.1: There are many NADH regeneration systems available. How would the thermodynamic compare to a system, with NADH (instead of DTT), where the cofactor is regenerated (alcohol dehydrogenase or glucose dehydrogenase). Also, would it be thermodynamically feasible to have an intrinsic NADH regeneration system (without DTT) with the methanol dehydrogenase, if the methanol concentration is high enough. Which methanol concentration would thermodynamically allow an intrinsic NADH regeneration and is this comparable to the alcohol oxidase system with DTT. Additionally, move the citation for the ΔG calculation (eQuilibrator) to the Fig1 caption, otherwise its not clear how these values were obtained.

Line 125: [...] formate reduction and methanol dehydrogenation [...]

Fig. 2: You might want to add O₂  H₂O₂ for the reaction of the alcohol oxidase. And I found it misleading, that there is also an arrow from CO₂ to methanol, since you do not show the reduction of CO₂ to methanol in your study. Maybe already a dashed arrow would clarify it.

Line 194: [...] similarity (57%) (Supplementary Fig. 3) [...]

Fig. 3: In panel a show also C951

Line 209: [...] and modeled the oxidized [...] – The word “simulated” is misleading, by suggesting a MD or QMMM simulation.

Line 304: Mention which specific mutation of the H protein was used in these experiments.

Line 333: Which bicarbonate concentration was eventually used in the assay?

Line 384: Over which time period is the 77.8 nmol/min/mg considered? Note that the rGPS-MCG [citation nr 16] pathway reports 100 nmol/min/mg over the initial period of the first 1.5h.(see also below for the comments on Table 1). Also discuss this number in respect to the C1 origin from methanol or CO₂. In your pathway the C3 compounds fix two C1 equivalents from the reduced methanol. The lower rates of other in vitro cycle certainly originate from the fact, that solely CO₂ as oxidized compound is incorporated and requires overall more energy and reducing power, than a mixture of CO₂ and reduced methanol.

Fig.6: In panel c add "THF 1mM" to the table in the plot. In d add also the dashed line at the 0.5 h mark.

Line 350: State in the Materials and Methods section where the AOX and CAT were obtained from.

Table 1:

For the CETCH row: correct the citation to "17"

For the ASAP row: correct the citation to "15"

For the rGPS-MCG row: correct the citation to "16"

For the ICE-CAP row: Add the concentration of the CO₂ in the "Air derived CO₂" statement. The 16mM should be replaced by 13.1 mM from line 352

General comments:

The production of serine (Fig.6 c) and pyruvate (Fig. 6 d) show quite different profiles. Could you argue, why the production of pyruvate ends faster and plateaus at around 7 mM, while the serine production seems to continue.

Please add the yields of your products (in the glycine, serine and pyruvate production from air derived CO₂) in correlation to the theoretically possible yield. If I am correct, you achieved around 65 % in the glycine production from 20 mM methanol (if CO₂ and NH₄ is not limiting). For the serine and pyruvate production you achieved around 30% of the theoretical yield from 50mM methanol (if CO₂ and NH₄ is not limiting) and taking into account that the methanol serves twice as methyl-donor. Could you comment on, why the conversion dropped so drastically even though only one respectively two enzymes were added.

Do you detect any side products during the production, which would partially explain the "incomplete" conversion? If yes, could you also comment on their origin.

Is the ratio of T-protein and SHMT important for the C3 compound production since they compete for the same substrate (5,10-CH₂-THF)?

Supplementary Figures 2, 3, 5, 6, 8 and 12 are not quoted in the manuscript text.

Reviewer #3 (Remarks to the Author):

Liu et al reported an ATP and NAD(P)H-independent catalytic system that converts methanol and CO₂ into the C₂ amino acid glycine. This is an interesting study with delicate design, and the key of the design is to use dithiothreitol (DTT) as a reducing agent to replace the reductive reaction which is generally catalyzed by the NAD(P)H-dependent L protein, thus creating thermodynamic driving force for the reversible glycine cleavage system (rGCS) to proceed towards the synthesis of glycine. The results obtained in this manuscript, if firmly demonstrated, would be a nice addition to this field.

The major concern for this manuscript is the lack of ¹³C labelled experiment to confirm the carbon fixation, as well as inappropriate use of the protein amount for calculating the CO₂ fixation rate. Since CO₂ is not the sole carbon source used in this study (methanol and formaldehyde were also used), it is necessary to use ¹³C labelled CO₂ to determine how much CO₂ was indeed incorporated into the final product. This is also a standard requirement for CO₂ fixation research as did in many previous ones. In addition, the CO₂ fixation rate from methanol and CO₂ was only calculated based on the amount of rGCS, which is incorrect as the AOX and catalase used for converting methanol into formaldehyde are essential for the overall reaction thus cannot be neglected. In this connection, the novel conclusion of this manuscript cannot be fully supported by the current available data. Moreover, since the reaction involves methanol and formaldehyde, it is misleading if the title does not include these organic C₁ compounds.

The reviewer also suggested other experiments in the specific comments that may help the authors to strengthen this story.

Specific comments below:

- Line 57-58: when talking about photosynthesis, the energy from sunlight cannot be neglected. Without the energy input, CO₂ cannot be fixed.

- Line 58-59: when talking about "Technical biosynthesis using gaseous and air-derived CO₂", one must think about where is the required energy coming from.

- Line 63-64: CO₂ cannot be considered as carbon resource equivalent to fossil, simply because CO₂ does not contain energy thus is actually a waste rather than a resource. In the absence of energy input, it is inappropriate to say that CO₂ can be used to replace fossil carbon.

- As shown in equation [6-3'] in Figure 1, DTT will be oxidized, and if not reduced it will need to be added to the reaction system continuously (this can also be seen in Figure 2a). In addition, the 5,10-CH₂-THF and THF appear as reactant and product in equation [6-3'] – if it means 5,10-CH₂-THF needs to be constantly added to the reaction system, it would be more complex than using ATP and NADPH. However from Figure 2a the 5,10-CH₂-THF and THF are well recycled. Please explain why 5,10-CH₂-THF

and THF appear as reactant and product in equation [6-3'], and whether DTT needs to be constantly added to drive the reaction.

- Line 126-127: "the large negative $\Delta rG'$ value in pathway 6, especially for the first reaction step, indicates a strong driving force in transforming methanol into glycine." The oxidation of methanol into formaldehyde is known to have large negative $\Delta rG'$ value, this only indicate a strong driving force from methanol to formaldehyde. The driving force from methanol to glycine is actually decided by the last step. It can be learned from the manuscript that the $\Delta rG'$ value of reaction [6-3'] (-8.0 KJ/mol) is lower than that of reaction [6-3] (-1.2 KJ/mol), and this slight decrease in $\Delta rG'$ value makes the conversion of 5,10-CH₂-THF into glycine much easier. Please clarify the relevant expression.

- Figure 2a: CO₂ will not be automatically converted into methanol, energy must be invested – therefore Figure 2a must be revised – the present figure shows two molecules of CO₂ will be converted into methanol without indication of energy or reducing power input.

- Figure 2d: please indicate the concentration all reactants used, as indicated in the table of Figure 2e. It is nice to learn that DTT consumption can be quantified – please indicate in Figure 2d what's the initial concentration of DTT and what is the final concentration of DTT after 10 min (600 s) reaction.

- Figure 2e: please include the time-course of the reaction. How long did the reaction last? Is the glycine production rate the maximum rate or the average rate?

- Figure 2e: it is remarkable to see the glycine formation rate was increased from 0.12 $\mu\text{M}/\text{min}$ in the presence of L and NADH, to 1.88 $\mu\text{M}/\text{min}$ with TCEP and 6.44 $\mu\text{M}/\text{min}$ with DTT. Please calculate the turnover number of the rGCS system.

- Line 180-183: The Supplementary Fig. 4 shows L protein is used in the unit of μmol , while DTT is used in the unit of mmol. This should be noted in the text. That is to say, it is not fair to compare the absolute rate without considering the amount of the reducing agent used, as the authors did in the text. The reviewer understands it might be difficult to increase the L protein into mmol concentration. The reaction rate should be tested if only μmol level DTT is used in the reaction (or at a concentration equivalent to that of the L protein).

- Figure 3c: mutation C335S should in principle also break the disulfide bond, why its glycine formation rate decreased in the absence of DTT? Addition of C335S significantly increased the glycine formation rate, indicating the disulfide bond somewhere else was reduced by DTT – as C335S cannot form disulfide bond. Please clarify.

- Figure 3d: Moreover, the addition of DTT contributed mostly to the WT in terms of the glycine formation rate. If the disulfide bond theory that the authors proposed is correct, then one would expect to see further increase in glycine formation rate by C949S or C951S or C949S/C951S. However this is not

the case. Please clarify.

- Line 226-228: "The disulfide bridge between C335 and C949 (C951) is reduced with DTT and the free C terminus is folded away from the active site entrance, leading to enhanced glycine synthesis" – this seems a hypothesis not proven. However, it looks like a conclusive statement but this statement cannot be supported by the data present in Figure 3c or 3d. If the authors want to prove their theory for the redox state of P protein, the reduced- and oxidized-state of P protein of WT and all mutants should be analysed and presented, in correlation with Figure 3c and 3d.

Line 231-233: "They form a positively charged surface close to the active site entrance, which is a beneficial driving force to absorb negatively charged bicarbonate." Again, this is a hypothesis driven from electrostatic surface analysis, but the authors do not have experimental data to support it.

- Supplementary Figure 7: the readers might want to know what is the effect of those mutants (C335S, C949S, C951S and C949S/C951S) on the electrostatic surface analysis of the oxidized state and reduced state of P protein.

- Figure 3: Are the conditions for glycine production the same as shown in Figure 2e? If not, please specify in the legend of Figure 3.

- The title of this section is "enhanced carboxylase activity of P protein through the change of redox state" – however, there is no data on the carboxylase activity of P protein. Glycine production rate cannot be simply used to indicate the carboxylase activity of P protein. The current data can only support a title "enhanced production of glycine by adding DTT".

- Overall, the reviewer does agree that addition of DTT promoted the glycine formation rate, but the title of this section "Enhanced carboxylase activity of P protein through the change of redox state" is not convincing due to the questions listed above.

- The observation of the polymerized form of P protein is very interesting.

(1) Please indicate the DTT concentration added in the experiments shown in Figure 3f.

(2) Why there is 2P, 4P, 5P, 6P, but not 3P?

(3) The authors argued that depolymerization of P protein in the presence of DTT contributed to the increased glycine formation rate. In this connection, the native PAGE of C335S, C949S, and C951S variant in the presence and absence of DTT should be presented to check whether it has any correlation with Figure 3c and Figure 3d. In another words, the intensity of P+DTT shown in Figure 3f was slightly stronger than that of C949S and C951S in the absence of DTT, but the glycine formation rate of P+DTT (Figure 3d) was 20 times higher than that of C949S and C951S in the absence of DTT (Figure 3c). Could the authors explain?

(4) In the case of T protein, why addition of DTT will convert 4T and 2T into 3T?

- Figure 4d: the molecular dynamics simulation of D68A should be included as a negative control. For supplementary Figure 9: the molecular dynamics simulation of Y70C, Y70M and Y70Q should be included as negative control.

- Figure 5e: please indicate HCO₃⁻ concentration after taking up gaseous CO₂ using KOH.

- Please use ¹³C labelled bicarbonate to perform an experiment to quantify exactly how much ¹³C labelled CO₂ was incorporated into glycine. This is a standard requirement in CO₂ fixation research, as shown in the CETCH cycle.

- Line 316-317: The authors stated “we obtained 15.5 mM (1.2 g/L) glycine in 3.5 h with a high carbon yield of 86% based on the consumed formaldehyde”. Since the subject of this manuscript is “CO₂ fixation”, it is inappropriate to use the yield based on formaldehyde. That’s why ¹³C labelled experiment is needed as it is very important to quantify exactly how much CO₂ was fixed in this experiment setup.

- Line 333: How was the 200 mM potassium bicarbonate added to the reaction system? Please indicate the actual potassium bicarbonate concentration used in figure 5e. How was the pH controlled? Why formaldehyde was added twice in figure 5e but added once in figure 5c?

- Figure 6c: The added 50 mM methanol was nearly consumed (residual concentration approx. 2 mM); the residual HCHO concentration approx. 3 mM; the residual glycine concentration approx. 2 mM. Let’s say 40 mM methanol was converted into serine, theoretically it should generate approx. 13 mM. The actual serine concentration is less than 8 mM. Please discuss the possible reason.

- As illustrated by the authors, the MsLSD was very active. Why pyruvate stopped increasing after 5 h reaction, even if there is still serine present? (figure 6d).

- Why there is no dashed line in figure 6d as shown in figure 6c?

- Again, ¹³C labeled bicarbonate are required to be used to check how much ¹³C labelled serine and pyruvate were produced. The authors can use the optimized reaction conditions to carry out such experiments.

- Line 384: the glycine production rate based on the rGCS protein is inappropriate. The amount of AOX and catalase used should be counted as the authors started the calculation from methanol. The reason is the oxidation from methanol to formaldehyde cannot proceed without AOX and catalase. Methanol is an energy-intensive substrate, but there is no free lunch. In the ASAP story, the starch synthesis rate, 22 nmol/mg/min, was calculated by using total amount of both catalyst and proteins. In the ASAP article, the CO₂ fixation rate of CETCH was recalculated to be 3.87 nmol/mg/min based on the amount of total proteins used in CETCH cycle.

Point by Point Reply to the comments of reviewers.

REVIEWER COMMENTS

Reviewer #1 (Remarks to the Author):

Based on the reductive glycine pathway, the author demonstrated a cell-free, ATP and NADH-independent CO₂ fixing system to produce glycine and serine from captured CO₂ (bicarbonate) and methanol. To attain NADH-independent CO₂ fixation, they replaced NADH with DTT to provide a stronger reducing power for glycine cleavage system (GCS). The results showed that DTT can improve the reduction rate of the H protein and simultaneously elevate carboxylase activity of the P protein through reducing disulfide bonds. Nevertheless, the kinetic activity of the wild-type H protein is not high enough for efficient glycine production. As a result, they engineered the lipoamide arm of H protein to generate a Y70A variant with a 2-fold increase in activity. Implementing the engineered protein, the DTT driven enzymatic system can convert bicarbonate and methanol into glycine at a rate of 77.8 nmol/min/mg, which is the highest CO₂ fixation rate to date among reported cell-free systems.

➤ Thanks for the feedback.

Major comments:

1. Although the authors claimed their cell-free system achieved the highest CO₂ fixation rate reported to date, the novelty and scientific insight of this work is limited. Fundamentally, the research team demonstrated an enzymatic reaction of reductive glycine pathway that can be driven by DTT in vitro. Using a chemical reducing agent to serve as an electron source is not sustainable. The problem cofactor regeneration is now shifted to DTT.

➤ Thanks for the feedback. First, we would like to highlight that there are several unique advantages to combining biocompatible DTT-based chemistry with enzymatic chemistry. The regeneration of DTT is discussed in the answer to the second question.

1) Using chemistry to boost thermodynamic limitations in biology

The replacement of L-protein and NADH by using DTT can decrease the Δ_rG' of the whole rGCS system from -1.2 kJ/mol to -8.0 kJ/mol (Fig. 1), indicating a much higher thermodynamic driving force in the presence of DTT.

2) Turning the reversible GCS system into the irreversible glycine-synthesis direction

The GCS system, normally comprised of H, P, T and L proteins, catalyzes fully reversible reactions in terms of glycine cleavage or synthesis^{1,2}. However, when L protein is replaced by DTT, the unidirectional electron flow from DTT to Hox, which is determined by the reduction potential, decides the predominant glycine

synthesis direction.

3) Substituting NAD(P)H and L protein

NAD(P)H is an expensive cofactor (2 \$/g NADH, 100 \$/g NADPH in large scale), even though it can be normally regenerated in the system. The initial concentration of NAD(P)H used in the cell-free enzymatic reactions is substantial, for example, 4 mM NADPH in the CETCH cycle and 5 mM NADPH in the POAP cycle. The regeneration of NAD(P)H by formate dehydrogenase, alcohol dehydrogenase or other enzymes also increases the cost. However, DTT is much cheaper (0.08 \$/g DTT in large scale) and it is feasible to regenerate DTT using electrochemical methods.

4) Enhancing glycine production rate kinetically

In the presence of DTT, the glycine synthesis rate is improved by more than 50-fold compared to the condition without DTT (Fig. 3). DTT can avoid the formation of P protein multimers and enhance its activities significantly.

2. Efficient cofactor regeneration in a sustainable manner is critical for a cell-free system to fix CO₂ at scale. 1) Can the authors discuss possible potential ways to transfer electron from sustainable source to DTT? 2) Also is DTT stable enough to be used in the cell-free system aerobically?

➤ 1) DTT can be regenerated using electrochemical methods and we discussed the potential ways to transfer electrons to DTT electrochemically in the “discussion” part. It has been demonstrated that the oxidized DTT (DTT_{ox}) can get electrons to regenerate DTT_{red} from a tungsten cathode in neutral aqueous solutions at a potential of -1.0V vs. SCE and this DTT_{red} in turn reduce lipoamide by thiol disulfide interchange^{3,4}. During this process, DTT serves as an excellent electron mediator.

2) We tested the stability of DTT and the new data is supplemented in Supplementary Fig. 16. Under environmental conditions, only approximately 0.5 mM DTT is oxidized from the initial 40 mM DTT in 8 hours.

Fig. 16 Measurement of DTT stabilities under conditions of the reaction system. a. DTT oxidation

over time. 40 mM DTT was freshly prepared by adding DTT into 50 mM Tris-HCl (pH=7.5) buffer and the solution was placed under 37°C in the presence of air. We took samples for scanning between 240 nm and 400 nm. b. DTT_{ox} has a specific absorbance at 280 nm.

3. The title of this manuscript and the claims at Line 109 are misleading. Their system cannot directly use CO₂ as sole feedstock for amino acids production.

➤ We removed the word “direct” and changed the title to “Turn gaseous and air-captured CO₂ into amino acid and pyruvate in an ATP and NAD(P)H-free chemoenzymatic system”. We also rephrased some sentences, including the claims at line 109.

4. The use of term “air-derived CO₂” may be imprecise. More importantly, emphasizing on the direct use of “air-derived CO₂” in this article is not appropriate. They performed chemical absorption to capture CO₂ instead of direct air capture by enzymes, and the chemical reaction unit was totally separated from the enzymatic module.

➤ We used CO₂ captured from air, not from pure CO₂ gas or from effluent or flue gas. We have now changed “air-derived” to “air-captured” to emphasize this fact. We have rephrased some other sentences in the text, including removing the word of “direct use” in the whole article, discussing the energy input used to capture CO₂.

Minor comments:

1. Line 102: The statement of “The latter is able to enhance the thermodynamic driving force by breaking the bio-thermodynamic boundary.....” is unclear.

➤ We have rephrased this sentence by referring to thermodynamic data to make it more clear.

2. Fig. 6a: The diagram in the middle of figure is confusing. C₁ chemicals cannot be converted to C₃ chemicals directly in their work.

➤ We have corrected it.

3. The authors can conduct ¹³C-labelling experiments to support that all the amino acids were derived from C₁ feedstock.

➤ We performed ¹³C-labelling experiments, including H¹³CO₃⁻, H¹³CHO, and H¹³CO₃⁻+ H¹³CHO labelling experiments. The new data is included in the updated Fig. 5 and supplementary Fig. 17, 18, 21. All the data support that the two carbons of glycine (-CH₂, -COOH) are derived from HCHO and HCO₃⁻, individually. The mole ratio of glycine to CO₂ is 1:1.

Supplementary Fig. 17 The mass spectrogram of glycine 2TBDMS (tert-Butyldimethylsilyl) derivative under different environmental conditions. We carried out NaH¹³CO₃, H¹³CHO, and NaH¹³CO₃ + H¹³CHO labelling experiments, respectively. The carbon donor in the reaction system containing Tris-HCl (50 mM, pH 7.5), 20 mM DTT, 0.5 mM THF, 10 mM HCHO, 50 mM NaHCO₃, 100 mM NH₄Cl, 25 μM PLP, 5 μM P-protein, 5 μM T-protein, and 60 μM Hox, was replaced by the corresponding 10 mM H¹³CHO, 50 mM NaH¹³CO₃ and their combinations.

Supplementary Fig. 18 Relative abundance of glycine mass isotopomer with ¹³C-labelling HCO₃⁻ or (and) HCHO as the carbon donor. a, m/z=218. b, m/z=246.

Supplementary Fig. 21 The mass spectrogram of serine 2TBDMS derivative to confirm that ^{13}C labelled $H^{13}CHO$ is added in the β -C of serine. The mole ratio of serine to CO_2 is 1:1.

Reviewer #2 (Remarks to the Author):

The manuscript describes the in vitro application of the reversible glycine cleavage system (rGCS) for the conversion of the C1 compounds CO_2 and methanol into the C2 compound glycine (as well as C3 compounds serine and pyruvate). Remarkably, the central pathway consists of only 4 core enzymes and achieves the conversion of abundant methanol, CO_2 and ammonia into an amino acid at the dispense of dithiothreitol (DTT) with high conversions. The study compares the rGCS pathway to other CO_2 converting in vitro pathways. The study thoroughly investigates the effect of DTT on the reduction of the lipoic acid as well as its secondary regulatory effect on the enzymes involved. Overall, the study shows in the beginning, how an in vitro pathway with the alcohol oxidase and the DTT dependent rGCS pathway shows favorable thermodynamics in comparison to natural occurring in vivo options. It then pursues to investigate and implement the in vitro pathway with satisfying results. Bottlenecks are removed very methodically by optimizing reaction conditions and apply enzyme engineering. It represents an elegant example in the emerging field of in vitro biosynthesis and underlines the capability of chemoenzymatic processes.

➤ Thank you for your positive comments.

Specific comments:

Fig.1: There are many NADH regeneration systems available. A. How would the thermodynamic compare to a system, with NADH (instead of DTT), where the cofactor

is regenerated (alcohol dehydrogenase or glucose dehydrogenase). B. Also, would it be thermodynamically feasible to have an intrinsic NADH regeneration system (without DTT) with the methanol dehydrogenase, if the methanol concentration is high enough. Which methanol concentration would thermodynamically allow an intrinsic NADH regeneration and is this comparable to the alcohol oxidase system with DTT. C. Additionally, move the citation for the $\Delta_r G'$ calculation (eQuilibrator) to the Fig 1 caption, otherwise its not clear how these values were obtained.

➤ Thanks for the good points.

A. It will increase the thermodynamic driving force with a cofactor regeneration system theoretically. For example:

Using DTT in our system is not only beneficial for a higher thermodynamic driving force, but it offers advantages in several other aspects:

1) Turning the reversible GCS system into the irreversible glycine-synthesis direction

The GCS system, normally comprised of H, P, T and L proteins, catalyzes fully reversible reactions in terms of glycine cleavage or synthesis^{1,2}. However, when L protein is replaced by DTT, the unidirectional electron flow from DTT to Hox, which is determined by the reduction potential, decides the predominant glycine synthesis direction.

2) Substituting NAD(P)H and L protein in a much cheaper way

NAD(P)H is an expensive cofactor (2 \$/g NADH, 100 \$/g NADPH in large scale), even though it can be normally regenerated in the system. The initial concentration of NAD(P)H used in the cell-free enzymatic reactions is substantial, for example, 4 mM NADPH in the CETCH cycle and 5 mM NADPH in the POAP cycle. The regeneration of NAD(P)H by formate dehydrogenase, alcohol dehydrogenase or other enzymes also increases the cost. However, DTT is much cheaper (0.08 \$/g DTT in large scale) and it is feasible to regenerate DTT using electrochemical methods.

3) Enhancing glycine production rate kinetically

In the presence of DTT, the glycine synthesis rate is improved by more than 50-folds compared to the condition without DTT (Fig. 3). DTT can destroy the formation of P protein multimers and enhance its activities significantly.

B. It is not thermodynamically feasible to have an intrinsic NADH regeneration system using methanol dehydrogenase, because of a high barrier to turn methanol to formaldehyde.

To make it comparable to our alcohol oxidase and DTT system thermodynamically, the methanol concentration should be around 4000 mol/L!

C. We included the eQuilibrator and MetaCyc citations in the Fig. 1 caption.

Line 125: [...] formate reduction and methanol dehydrogenation [...]

➤ Corrected.

Fig. 2: You might want to add $O_2 \rightarrow H_2O_2$ for the reaction of the alcohol oxidase. And I found it misleading, that there is also an arrow from CO_2 to methanol, since you do not show the reduction of CO_2 to methanol in your study. Maybe already a dashed arrow would clarify it.

➤ Thanks for pointing it out. We have modified the Fig. 2a as suggested.

Line 194: [...] similarity (57%) (Supplementary Fig. 3) [...]

➤ Added.

Fig. 3: In panel show also C951

➤ Added.

Line 209: [...] and modeled the oxidized [...] – The word “simulated” is misleading, by suggesting a MD or QMMM simulation.

➤ “simulated” is changed to “modeled”.

Line 304: Mention which specific mutation of the H protein was used in these experiments.

➤ Added.

Line 333: Which bicarbonate concentration was eventually used in the assay?

➤ We added the concentration in the sentence. 50 mM.

Line 384: A. Over which time period is the 77.8 nmol/min/mg considered? Note that the rGPS-MCG [citation nr 16] pathway reports 100 nmol/min/mg over the initial period of the first 1.5h.(see also below for the comments on Table 1). B. Also discuss this number in respect to the C1 origin from methanol or CO_2 . In your pathway the C3 compounds fix two C1 equivalents from the reduced methanol. The lower rates of other in vitro cycle certainly originate from the fact, that solely CO_2 as oxidized compound is incorporate requires overall more energy and reducing power, than a mixture of CO_2 and reduced methanol.

➤ A. We recalculated the C1 fixation rate based on the total number of enzymes, including AOX, CAT and rGCS. The previous data was based on 11 mM glycine from 3 h on rGCS. Now the updated data of 53.7 nmol/min/mg is based on 5 h reaction period (13.2 mM glycine is formed from methanol and CO_2 in 5 h in Fig. 6b).

B. This is a good point. In other CO_2 fixation pathways as shown in Table 1, most of the CO_2 fixation pathways require an energy-rich compound as a co-substrate. For example, propionyl-CoA is required in the CETCH, acetate for the POAP, Crotonyl CoA and PEP for the rGPS-MCG. These energy-rich compounds are used to capture CO_2 by providing electrons and energy. But these co-substrates are difficult to produce and increase the difficulties as well as costs to the process. In our case, the use of C1 compound methanol (rich in energy and reducing power),

which can be obtained from CO₂ chemically or electrochemically, to fix CO₂ (double C1 fixation strategy) shows great potential in terms of efficient energy utilization and the development of sustainable process. We have clarified this in our “discussion” part.

Fig.6: In panel c add "THF 1mM" to the table in the plot. In d add also the dashed line at the 0.5 h mark.

➤ Added.

Line 350: State in the Materials and Methods section where the AOX and CAT were obtained from.

➤ Added.

Table 1:

For the CETCH row: correct the citation to "17"

For the ASAP row: correct the citation to "15"

For the rGPS-MCG row: correct the citation to "16"

For the ICE-CAP row: Add the concentration of the CO₂ in the “Air derived CO₂” statement. The 16mM should be replaced by 13.1 mM from line 352

➤ Thanks. We have corrected all of them.

General comments:

The production of serine (Fig.6 c) and pyruvate (Fig. 6 d) show quite different profiles. Could you argue, why the production of pyruvate ends faster and plateaus at around 7 mM, while the serine production seems to continue.

➤ This is because of the toxicity of formaldehyde on MsLSD activities.

For the serine and pyruvate production, we added 50 mM methanol in the reaction system, which was able to generate more than 20 mM formaldehyde during the reaction process. We tested the toxicity of formaldehyde on SHMT and MsLSD activities and the new data is added in the updated supplementary Fig. 24. It was found that MsLSD was quite sensitive to the concentrations of formaldehyde and 10 mM formaldehyde lost half of its activity, while 20 mM formaldehyde lost its 75% activity. SHMT is more robust and it can tolerate a high concentration of formaldehyde (the condensation of formaldehyde and THF is the substrate for SHMT). At 50 mM formaldehyde, the activity of SHMT maintained at 75% of its activity compared to that achieved at 10 mM formaldehyde.

Please add the yields of your products (in the glycine, serine and pyruvate production from air derived CO₂) in correlation to the theoretically possible yield. If I am correct, you achieved around 65 % in the glycine production from 20 mM methanol (if CO₂ and NH₄ is not limiting). For the serine and pyruvate production you achieved around 30% of the theoretical yield from 50mM methanol (if CO₂ and NH₄ is not limiting)

and taking into account that the methanol serves twice as methyl-donor. Could you comment on, why the conversion dropped so drastically even though only one respectively two enzymes were added.

➤ In the presence of 50 mM methanol, it generated more than 20 mM formaldehyde in the process. On one hand, a high concentration of formaldehyde is detrimental to the enzymes, including SHMT and LSD. This new result has been added in the supplementary Fig. 24; On the other hand, we noticed that when the concentration of formaldehyde is high (refer to more than 20 mM), it formed white precipitate through chemical reaction between formaldehyde and NH_4^+ . There was no precipitate when the concentration of formaldehyde is lower than 20 mM. This phenomenon has been observed in previous studies (Ref.33). We have clarified this in the main text.

Do you detect any side products during the production, which would partially explain the “incomplete” conversion? If yes, could you also comment on their origin.

➤ As answered in the last question, we noticed the formation of white precipitates. It is formed through a high concentration of formaldehyde and NH_4^+ .

Is the ratio of T-protein and SHMT important for the C3 compound production since they compete for the same substrate (5,10-CH₂-THF)?

➤ Good point. According to previous study (Ref.33), the concentration of T protein was around 5 μM for the best glycine synthesis. For the production of C3 compounds, we used a high concentration of SHMT at 20 μM , because this step is not thermodynamically feasible. It did not help for increased production of serine or pyruvate by furthering enhancing the concentration of SHMT.

Supplementary Figures 2, 3, 5, 6, 8 and 12 are not quoted in the manuscript text.

➤ We have added them in the main text.

Reviewer #3 (Remarks to the Author):

Liu et al reported an ATP and NAD(P)H-independent catalytic system that converts methanol and CO₂ into the C₂ amino acid glycine. This is an interesting study with delicate design, and the key of the design is to use dithiothreitol (DTT) as a reducing agent to replace the reductive reaction which is generally catalyzed by the NAD(P)H-dependent L protein, thus creating thermodynamic driving force for the reversible glycine cleavage system (rGCS) to proceed towards the synthesis of glycine. The results obtained in this manuscript, if firmly demonstrated, would be a nice addition to this field.

➤ Thanks for the positive feedback.

A. The major concern for this manuscript is the lack of ^{13}C labelled experiment to confirm the carbon fixation, as well as inappropriate use of the protein amount for calculating the CO_2 fixation rate. Since CO_2 is not the sole carbon source used in this study (methanol and formaldehyde were also used), it is necessary to use ^{13}C labelled CO_2 to determine how much CO_2 was indeed incorporated into the final product. This is also a standard requirement for CO_2 fixation research as did in many previous ones. B. In addition, the CO_2 fixation rate from methanol and CO_2 was only calculated based on the amount of rGCS, which is incorrect as the AOX and catalase used for converting methanol into formaldehyde are essential for the overall reaction thus cannot be neglected. In this connection, the novel conclusion of this manuscript cannot be fully supported by the current available data. C. Moreover, since the reaction involves methanol and formaldehyde, it is misleading if the title does not include these organic C1 compounds.

- A. We performed ^{13}C -labelling experiments, including $\text{H}^{13}\text{CO}_3^-$, H^{13}CHO , and $\text{H}^{13}\text{CO}_3^- + \text{H}^{13}\text{CHO}$ labelling experiments. The new data is included in the updated Fig. 5 and supplementary Fig. 17, 18. All the data support that the two carbons of glycine are derived from HCHO ($\alpha\text{-C}$) and HCO_3^- , individually. The mole ratio of glycine to CO_2 is 1:1.

Supplementary Fig. 17 The mass spectrogram of glycine 2TBDMS (tert-Butyldimethylsilyl) derivative under different environmental conditions. We carried out $\text{NaH}^{13}\text{CO}_3$, H^{13}CHO , and $\text{NaH}^{13}\text{CO}_3 + \text{H}^{13}\text{CHO}$ labelling experiments, respectively. The carbon donor in the reaction system containing Tris-HCl (50 mM, pH 7.5), 20 mM DTT, 0.5 mM THF, 10 mM HCHO , 50 mM NaHCO_3 , 100 mM NH_4Cl , 25 μM PLP, 5 μM P-protein, 5 μM T-protein, and 60 μM Hox, was replaced by the corresponding 10 mM H^{13}CHO , 50 mM $\text{NaH}^{13}\text{CO}_3$ and their combinations.

Supplementary Fig. 18 Relative abundance of glycine mass isotopomer with ¹³C-labelling HCO₃⁻ or (and) HCHO as the carbon donor. a, m/z=218. b, m/z=246.

B. We recalculated the C1 fixation rate based on the total number of enzymes, including AOX, CAT and rGCS. The updated data of 74.6 nmol/min/mg is based on 3 h reaction period (11 mM glycine is formed from methanol in 3 h).

C. Thanks for the point. We changed the title to “Turn gaseous and air-captured CO₂ into amino acid and pyruvate in an ATP and NAD(P)H-free chemoenzymatic system”. We did not include methanol or formaldehyde in the title because of several reasons: 1) Methanol can be derived from CO₂. There are many studies presenting methods on this conversion and a recent review by Navarro-Jaen⁵ summarized the strategies and challenges to reduce CO₂ to methanol, including direct use of air for methanol synthesis (Ref. 6, 29). 2) A similar article presented the co-consumption of CO₂ and methanol (providing energy)⁵ and they did not include methanol in the title. 3) It would be a bit longer to have “methanol or formaldehyde” in the title.

The reviewer also suggested other experiments in the specific comments that may help the authors to strengthen this story.

Specific comments below:

- Line 57-58: when talking about photosynthesis, the energy from sunlight cannot be neglected. Without the energy input, CO₂ cannot be fixed.

➤ We have rephrased this sentence.

- Line 58-59: when talking about “Technical biosynthesis using gaseous and air-derived CO₂”, one must think about where is the required energy coming from.

➤ We emphasized the use of CO₂ as a carbon source. The sustainable energy input is an important factor in CO₂ fixation process. We have clarified this energy part in the updated “discussion”. In most of CO₂ fixation pathways as shown in Table 1, they require an energy-rich compound as a co-substrate. For example, propionyl-CoA is required in the CETCH, acetate for the POAP, Crotonyl CoA and PEP for

the rGPS-MCG. These energy-rich compounds are used to capture CO₂ by providing electrons and energy. But these co-substrates are difficult to produce and increase the difficulties as well as costs to the process. In our case, the use of C1 compound methanol (rich in energy and reducing power), which can be obtained from CO₂ chemically or electrochemically, to fix electron deficient CO₂ (double C1 fixation strategy) shows great potential in terms of efficient energy utilization and the development of sustainable process.

- Line 63-64: CO₂ cannot be considered as carbon resource equivalent to fossil, simply because CO₂ does not contain energy thus is actually a waste rather than a resource. In the absence of energy input, it is inappropriate to say that CO₂ can be used to replace fossil carbon.

➤ Good point. The sustainable energy input is an important factor in CO₂ fixation process. We have clarified this energy part in the updated “discussion”, especially the unique advantage of our strategy to use electron intensive C1 compound methanol to capture electron deficient CO₂.

- As shown in equation [6-3'] in Figure 1, DTT will be oxidized, and if not reduced it will need to be added to the reaction system continuously (this can also be seen in Figure 2a). In addition, the 5,10-CH₂-THF and THF appear as reactant and product in equation [6-3'] – if it means 5,10-CH₂-THF needs to be constantly added to the reaction system, it would be more complex than using ATP and NADPH. However from Figure 2a the 5,10-CH₂-THF and THF are well recycled. Please explain why 5,10-CH₂-THF and THF appear as reactant and product in equation [6-3'], and whether DTT needs to be constantly added to drive the reaction.

➤ In our reaction system, DTT is oxidized continuously so that we added a high concentration of 20 mM DTT in the system. We did not add DTT constantly during the process. In future studies DTT can be regenerated using electrochemical methods and we have now discussed DTT regeneration. It should be mentioned that DTT is a cheap compound.

As the reviewer pointed out, THF and 5,10-CH₂-THF are well recycled. So we only added a low concentration of 0.5 mM THF in the reaction system. However, for thermodynamic analysis, we manually separated the system into three parts: [6-1] methanol oxidation to formaldehyde; [6-2] spontaneous condensation between formaldehyde and THF to generate 5,10-CH₂-THF; [6-3] glycine synthesis from the rGCS system using 5,10-CH₂-THF, NH₃ and CO₂. From the thermodynamic analysis (Fig.1), it is known that the reaction [6-3] is the bottleneck. Also the separation of the whole pathway into three parts can clearly demonstrate the role of DTT to replace NADH in benefiting the rGCS step.

- Line 126-127: “the large negative $\Delta rG'$ value in pathway 6, especially for the first reaction step, indicates a strong driving force in transforming methanol into glycine.” The oxidation of methanol into formaldehyde is known to have large negative $\Delta rG'$ value, this only indicate a strong driving force from methanol to formaldehyde. The

driving force from methanol to glycine is actually decided by the last step. It can be learned from the manuscript that the $\Delta rG'$ value of reaction [6-3'] (-8.0 KJ/mol) is lower than that of reaction [6-3] (-1.2 KJ/mol), and this slight decrease in $\Delta rG'$ value makes the conversion of 5,10-CH₂-THF into glycine much easier. Please clarify the relevant expression.

➤ Thanks for the point. We have clarified the relevant expressions in the first part.

- Figure 2a: CO₂ will not be automatically converted into methanol, energy must be invested – therefore Figure 2a must be revised – the present figure shows two molecules of CO₂ will be converted into methanol without indication of energy or reducing power input.

➤ We have revised Fig. 2a. The reduction of CO₂ to methanol has been indicated in a dashed line and the energy part is clarified in the “discussion” part. There are many studies presenting methods on this conversion and a recent review by Navarro-Jaen⁶ summarized the strategies and challenges to reduce CO₂ to methanol, including direct use of air for methanol synthesis (Ref. 6, 29). We did not work on the CO₂ to methanol part in our work.

- Figure 2d: please indicate the concentration all reactants used, as indicated in the table of Figure 2e. It is nice to learn that DTT consumption can be quantified – please indicate in Figure 2d what's the initial concentration of DTT and what is the final concentration of DTT after 10 min (600 s) reaction.

➤ We have added the concentrations of all reactants in the updated Fig. 2d. The initial concentration of DTT is 20 mM, because the concentration of H_{ox} was only 10 μM and the consumed DTT was very small in the beginning of 10 mins, it is beyond the detection of Ellman method.

- Figure 2e: please include the time-course of the reaction. How long did the reaction last? Is the glycine production rate the maximum rate or the average rate?

➤ We have updated Fig. 2e. The reaction was carried out for 1 h and the glycine production rate is the average rate of glycine production in 1 h.

- Figure 2e: it is remarkable to see the glycine formation rate was increased from 0.12 μM/min in the presence of L and NADH, to 1.88 μM/min with TCEP and 6.44 μM/min with DTT. Please calculate the turnover number of the rGCS system.

➤ We have calculated the turnover number of the rGCS system under different conditions and the data is 0.32 min⁻¹ for DTT and 0.095 min⁻¹ for TCEP and 0.005 min⁻¹ for L protein.

- Line 180-183: The Supplementary Fig. 4 shows L protein is used in the unit of μmol, while DTT is used in the unit of mmol. This should be noted in the text. That is to say, it is not fair to compare the absolute rate without considering the amount of the reducing agent used, as the authors did in the text. The reviewer understands it might be difficult to increase the L protein into mmol concentration. The reaction rate should be tested if

only μM level DTT is used in the reaction (or at a concentration equivalent to that of the L protein).

➤ Good point. We have made it clear in the text that DTT is added in the system at mM level, while L protein used is at μM level. We did two additional experiments to compare the L protein and DTT at the same concentration level. 1. Reduce the concentration of DTT to $5 \mu\text{M}$. 2. Increase the concentration of L protein to 1 mM level. The new results are included in the updated supplementary Fig.4. At $5 \mu\text{M}$ DTT, we did not observe detectable concentration of glycine. When the concentration of L protein was increased to 1 mM, the glycine production is around the same level to that achieved at a low concentration of L protein ($5 \mu\text{M}$), because the limiting step in rGCS lies in the carboxylation step catalyzed by P protein. So there are several unique roles that DTT plays in our system.

1) Using chemistry to boost thermodynamic limitations in biology

The replacement of L-protein and NADH by using DTT can decrease the $\Delta rG'$ of the whole rGCS system from -1.2 kJ/mol to -8.0 kJ/mol (Fig. 1), meaning a much higher thermodynamic driving force in the presence of DTT.

2) Turning the reversible GCS system into the irreversible glycine-synthesis direction

The GCS system, normally comprised of H, P, T and L proteins, catalyzes fully reversible reactions in terms of glycine cleavage or synthesis 1,2. However, when L protein is replaced by DTT, the unidirectional electron flow from DTT to Hox, which is determined by the reduction potential, decides the predominant glycine synthesis direction.

3) Substituting NAD(P)H and L protein

NAD(P)H is an expensive cofactor ($2 \text{ \$/g NADH}$, $100 \text{ \$/g NADPH}$ in large scale), even though it can be normally regenerated in the system. The initial concentration of NAD(P)H used in the cell-free enzymatic reactions is substantial, for example, 4 mM NADPH in the CETCH cycle and 5 mM NADPH in the POAP cycle. The regeneration of NAD(P)H by formate dehydrogenase, alcohol dehydrogenase or other enzymes also increases the cost. However, DTT is much cheaper ($0.08 \text{ \$/g DTT}$ in large scale) and it is feasible to regenerate DTT using electrochemical methods.

4) Enhancing glycine production rate kinetically

In the presence of DTT, the glycine synthesis rate is improved by more than 50-fold compared to the condition without DTT (Fig. 3). DTT can avoid the formation of P protein multimers and enhance its activities significantly.

- Figure 3c: mutation C335S should in principle also break the disulfide bond, why its glycine formation rate decreased in the absence of DTT? Addition of C335S significantly increased the glycine formation rate, indicating the disulfide bond somewhere else was reduced by DTT – as C335S cannot form disulfide bond. Please clarify.

➤ Because the C335 residue is located in the active site, its mutation will largely

destroy the activity of P protein (Ref. 21). It is correct that the addition of DTT to C335S increased its glycine formation rate, which indicates that in addition to the disulfide bridge between C335 and C949, some other disulfide bonds could be reduced, such as the disulfide bonds responsible for the formation of P polymers. This observation is in accordance with the results achieved by Hasse et al., 2013 (Ref. 21). The redox-dependent regulation of P activity is complex. We have clarified this in the updated main text.

- Figure 3d: Moreover, the addition of DTT contributed mostly to the WT in terms of the glycine formation rate. If the disulfide bond theory that the authors proposed is correct, then one would expect to see further increase in glycine formation rate by C949S or C951S or C949S/C951S. However this is not the case. Please clarify.

➤ As mentioned in the last question, the redox-dependent regulation of P protein is complex. It was demonstrated in the study of Hasse. et al., 2013 that when the disulfide bond was reduced between C335 and C949, the cofactor PLP can get into the active center of P protein. After that, P protein adopts conformational changes, such as C terminus extension that may interact with H protein. It is likely that the mutation of C949S or (and) C951S influences the mobility of the C terminus and that this may in turn influence substrate binding and catalysis. We have discussed this in the updated text.

- Line 226-228: “The disulfide bridge between C335 and C949 (C951) is reduced with DTT and the free C terminus is folded away from the active site entrance, leading to enhanced glycine synthesis” – this seems a hypothesis not proven. However, it looks like a conclusive statement but this statement cannot be supported by the data present in Figure 3c or 3d. If the authors want to prove their theory for the redox state of P protein, the reduced- and oxidized-state of P protein of WT and all mutants should be analysed and presented, in correlation with Figure 3c and 3d.

➤ The redox-dependent activity of P protein is presented by Hasse. et al., 2013. Hasse et al studied the redox regulation of cyanobacterial P protein (*Synechocystis* sp. PCC6803). The similarity of amino acids between this cyanobacterial P protein and *E. coli* P protein is approaching 60% (Supplementary Fig.3) and their 3D structures are almost the same. The data, including the crystallized structure of P_{ox} and P_{red} and its mutants from Hasse study confirmed that the disulfide bridge between C335 and C949 largely regulated P activity. To make it more clear, we have clarified that this conjecture is based on the reference of Hasse. et al., 2013.

Line 231-233: “They form a positively charged surface close to the active site entrance, which is a beneficial driving force to absorb negatively charged bicarbonate.” Again, this is a hypothesis driven from electrostatic surface analysis, but the authors do not have experimental data to support it.

➤ We have rephrased these sentences to make it clear that this is a hypothesis.

- Supplementary Figure 7: the readers might want to know what is the effect of those mutants (C335S, C949S, C951S and C949S/C951S) on the electrostatic surface analysis of the oxidized state and reduced state of P protein.

➤ We analyzed the electrostatic surface of these mutants and there are no clear differences on the electrostatic surface caused by the introduced single mutation.

- Figure 3: Are the conditions for glycine production the same as shown in Figure 2e? If not, please specify in the legend of Figure 3.

➤ Same.

- The title of this section is “enhanced carboxylase activity of P protein through the change of redox state” – however, there is no data on the carboxylase activity of P protein. Glycine production rate cannot be simply used to indicate the carboxylase activity of P protein. The current data can only support a title “enhanced production of glycine by adding DTT”.

➤ Good points. We intended to isolate the pure H_{int} as substrate and test the sole activity of P protein for glycine synthesis, but this experiment was failed due to the instability of H_{int} . So we tested the activity of P protein to convert H_{ox} to H_{int} on glycine cleavage with/without DTT and the data is presented in Fig. 6.

To make it clear, we agree to change the subtitle to “enhanced glycine synthesis through the change of redox state of P protein”.

- Overall, the reviewer does agree that addition of DTT promoted the glycine formation rate, but the title of this section “Enhanced carboxylase activity of P protein through the change of redox state” is not convincing due to the questions listed above.

- The observation of the polymerized form of P protein is very interesting.

(1) Please indicate the DTT concentration added in the experiments shown in Figure 3f.

➤ 50 mM

(2) Why there is 2P, 4P, 5P, 6P, but not 3P?

➤ It is difficult to identify the size of proteins accurately by native-page analysis.

It should be mentioned that the active P protein is a homodimer. 5P is probably due to the inherent shortcoming of native-page analysis.

(3) The authors argued that depolymerization of P protein in the presence of DTT contributed to the increased glycine formation rate. In this connection, the native PAGE of C335S, C949S, and C951S variant in the presence and absence of DTT should be presented to check whether it has any correlation with Figure 3c and Figure 3d. In another words, the intensity of P+DTT shown in Figure 3f was slightly stronger than that of C949S and C951S in the absence of DTT, but the glycine formation rate of P+DTT (Figure 3d) was 20 times higher than that of C949S and C951S in the absence of DTT (Figure 3c). Could the authors explain?

➤ The redox-dependent regulation of P protein is complex. It was demonstrated in the study of Hasse. et al., 2013 that when the disulfide bond was reduced between C335 and C949, the cofactor PLP can get into the active center of P protein. After that, P protein adopts conformational changes, such as C terminus extension that may interact with H protein. It is likely that the mutation of C949S or (and) C951S influences the mobility of the C terminus and that this may in turn influence substrate binding and catalysis. We have discussed this in the updated main text.

(4) In the case of T protein, why addition of DTT will convert 4T and 2T into 3T?

➤ We cannot completely explain the phenomenon at the moment. It should be mentioned that native page can normally not be used for accurate determination of molecular weight.

- Figure 4d: the molecular dynamics simulation of D68A should be included as a negative control. For supplementary Figure 9: the molecular dynamics simulation of Y70C, Y70M and Y70Q should be included as negative control.

➤ We have added the required negative controls supplementary Fig.12, 13 and discussed the results in the main text.

- Figure 5e: please indicate HCO₃⁻ concentration after taking up gaseous CO₂ using KOH.

➤ Added.

- Please use ¹³C labelled bicarbonate to perform an experiment to quantify exactly how much ¹³C labelled CO₂ was incorporated into glycine. This is a standard requirement in CO₂ fixation research, as shown in the CETCH cycle.

➤ We have conducted ¹³C labelled experiments. Please see the answer for the first question.

- Line 316-317: The authors stated “we obtained 15.5 mM (1.2 g/L) glycine in 3.5 h with a high carbon yield of 86% based on the consumed formaldehyde”. Since the subject of this manuscript is “CO₂ fixation”, it is inappropriate to use the yield based on formaldehyde. That’s why ¹³C labelled experiment is needed as it is very important to quantify exactly how much CO₂ was fixed in this experiment setup.

➤ We added ¹³C labelled experiments. We have calculated the glycine production yield based on bicarbonate and now included them in the main text.

- Line 333: How was the 200 mM potassium bicarbonate added to the reaction system? Please indicate the actual potassium bicarbonate concentration used in figure 5e. How was the pH controlled? Why formaldehyde was added twice in figure 5e but added once in figure 5c?

➤ It was added into the rGCS system at 50 mM as the carbon substrate. The reaction was carried out in 50 mM Tris buffer (pH=7.5). In Fig. 5c, we added 20 mM formaldehyde at the beginning of the reaction and observed a little white precipitate

that was caused by chemical reaction between a high concentration of formaldehyde (more than 20 mM) and NH_4^+ . This phenomenon has been observed in previous studies (Ref. 33). The yield from formaldehyde to glycine is about 78% of the theoretical maximum (Fig. 5C). But in Fig. 5e, we tested to add 20 mM formaldehyde in the beginning of the experiment and there was more white precipitates than that formed in Fig. 5c, so we added 10 mM formaldehyde in the beginning and another 10 mM formaldehyde after 1 h. We did not see clear precipitation under this condition.

- Figure 6c: The added 50 mM methanol was nearly consumed (residual concentration approx. 2 mM); the residual HCHO concentration approx. 3 mM; the residual glycine concentration approx. 2 mM. Let's say 40 mM methanol was converted into serine, theoretically it should generate approx. 13 mM. The actual serine concentration is less than 8 mM. Please discuss the possible reason.

➤ Thanks for pointing it out. In our reaction system, we noticed that when the concentration of formaldehyde is high (refer to more than 20 mM), it formed white precipitate through chemical reaction between formaldehyde and NH_4^+ . There was no precipitate when the concentration of formaldehyde is lower than 20 mM. This phenomenon has been observed in previous studies (Ref.33). This 'side-product' led to a decreased production yield. We have discussed this point in the updated main text.

- As illustrated by the authors, the MsLSD was very active. Why pyruvate stopped increasing after 5 h reaction, even if there is still serine present? (figure 6d).

➤ This phenomenon is caused by the toxicity of formaldehyde at high concentrations. For the serine and pyruvate production, we added 50 mM methanol in the reaction system, which was able to generate more than 20 mM formaldehyde during the reaction process. We tested the toxicity of formaldehyde on SHMT and MsLSD activities and the new data is added in the updated supplementary Fig. 110. It was found that MsLSD was quite sensitive to the concentrations of formaldehyde and 10 mM formaldehyde lost half of its activity, while 20 mM formaldehyde lost its 75% activity. SHMT is more robust and it can tolerate a high concentration of formaldehyde (the condensation of formaldehyde and THF is the substrate for SHMT). At 50 mM formaldehyde, the activity of SHMT maintained at 75%. To increase pyruvate production in future studies, one target is to enhance the tolerance of formaldehyde on LSD.

- Why there is no dashed line in figure 6d as shown in figure 6c?

➤ We apologize for the mistake and added the dashed line in Fig.6D.

- Again, ^{13}C labeled bicarbonate are required to be used to check how much ^{13}C labelled serine and pyruvate were produced. The authors can use the optimized reaction

conditions to carry out such experiments.

➤ We carried out ¹³C labeled experiments and confirmed the carbon source for glycine, serine and pyruvate. The new results are updated in Fig. 5 and supplementary Fig. 17, 18, 21.

- Line 384: the glycine production rate based on the rGCS protein is inappropriate. The amount of AOX and catalase used should be counted as the authors started the calculation from methanol. The reason is the oxidation from methanol to formaldehyde cannot proceed without AOX and catalase. Methanol is an energy-intensive substrate, but there is no free lunch. In the ASAP story, the starch synthesis rate, 22 nmol/mg/min, was calculated by using total amount of both catalyst and proteins. In the ASAP article, the CO₂ fixation rate of CETCH was recalculated to be 3.87 nmol/mg/min based on the amount of total proteins used in CETCH cycle.

➤ Thanks for the point. We recalculated the glycine production rate based on the total amount of proteins including AOX, CAT and the rGCS proteins. It is 53.7 nmol/min/mg based on the total consumption of C1 substrates. The previous data was based on 11 mM glycine from 3 h on rGCS. Now the updated data of 53.7 nmol/min/mg is based on 5 h reaction period (13.2 mM glycine is formed from methanol and CO₂ in 5 h in Fig. 6b).

References

1. Douce, R., Bourguignon, J., Neuburger, M. & Rébeillé, F. The glycine decarboxylase system: a fascinating complex. *Trends in Plant Science* **6**, 167–176 (2001).
2. Hong, Y., Ren, J., Zhang, X., Wang, W. & Zeng, A.-P. Quantitative analysis of glycine related metabolic pathways for one-carbon synthetic biology. *Current Opinion in Biotechnology* **64**, 70–78 (2020).
3. Wang, X. *et al.* Cofactor NAD(P)H Regeneration Inspired by Heterogeneous Pathways. *Chem* **2**, 621–654 (2017).
4. Shaked, Z., Barber, J. J. & Whitesides, G. M. Combined electrochemical/enzymic method for in situ regeneration of NADH based on cathodic reduction of cyclic disulfides. *J. Org. Chem.* **46**, 4100–4101 (1981).

5. Gassler, T. *et al.* The industrial yeast *Pichia pastoris* is converted from a heterotroph into an autotroph capable of growth on CO₂. *Nat Biotechnol* **38**, 210–216 (2020).
6. Navarro-Jaén, S. *et al.* Highlights and challenges in the selective reduction of carbon dioxide to methanol. *Nat Rev Chem* **5**, 564–579 (2021).

REVIEWERS' COMMENTS

Reviewer #1 (Remarks to the Author):

The authors have mostly answered my questions, except for the title change. As pointed out by reviewer #3 as well, the authors did not use only CO₂ as their substrate, but rather coupling methanol as well. AT LEAST methanol should be mentioned in the title, to avoid misleading.

Reviewer #2 (Remarks to the Author):

I appreciate the authors' efforts to address my previous comments and concerns. I am generally satisfied with all the answers and adaptations in the manuscript.

Reviewer #3 (Remarks to the Author):

The authors addressed most of the concerns raised by the reviewers. I have two more concerns:

1. The first question raised by Reviewer 1 was actually pointing out the essence of this study, which I agree. The authors reiterated 4 points which have been illustrated in the manuscript, however this 4 points did not fundamentally change the point raised by Reviewer 1. To my opinion, the authors did not address the point raised by Reviewer 1 – for instance – “Using a chemical reducing agent to serve as an electron source is not sustainable”. Actually Reviewer 1 is quite positive on the manuscript. He/she just illustrated some fundamental points that require the authors to address in the revised version. As the authors addressed in the following question – DTT can be regenerated using electrochemical method – then a new question comes, would the electrochemical regeneration of DTT can be compatible with the enzymatic reactions that the authors constructed?

2. Methanol needs to be incorporated in the title as they provides energy. Yes methanol can be derived from CO₂ at the expense of energy input (for instance H₂). But in the case of the authors, it would be misleading for the researchers if there is no methanol in the title – the reader would read the title as a novel process to turn CO₂ into organics without ATP and NADPH. However, the fact is that methanol has to be added to the process as a co-substrate, and methanol is the starting point for calculating the CO₂ fixation rate, therefore it cannot be neglected in the title.

Point by Point Reply to the comments of reviewers.

REVIEWERS' COMMENTS

Reviewer #1 (Remarks to the Author):

The authors have mostly answered my questions, except for the title change. As pointed out by reviewer #3 as well, the authors did not use only CO₂ as their substrate, but rather coupling methanol as well. AT LEAST methanol should be mentioned in the title, to avoid misleading.

➤ Thanks for the feedback.

We have changed the title to “Turn air-captured CO₂ with methanol into amino acid and pyruvate in an ATP/NAD(P)H-free chemoenzymatic system”.

Reviewer #2 (Remarks to the Author):

I appreciate the authors' efforts to address my previous comments and concerns. I am generally satisfied with all the answers and adaptations in the manuscript.

➤ Thanks.

Reviewer #3 (Remarks to the Author):

The authors addressed most of the concerns raised by the reviewers. I have two more concerns:

1. The first question raised by Reviewer 1 was actually pointing out the essence of this study, which I agree. The authors reiterated 4 points which have been illustrated in the manuscript, however this 4 points did not fundamentally change the point raised by Reviewer 1. To my opinion, the authors did not address the point raised by Reviewer 1 – for instance – “Using a chemical reducing agent to serve as an electron source is not sustainable”. Actually Reviewer 1 is quite positive on the manuscript. He/she just illustrated some fundamental points that require the authors to address in the revised version. As the authors addressed in the following question – DTT can be regenerated using electrochemical method – then a new question comes, would the electrochemical regeneration of DTT can be compatible with the enzymatic reactions that the authors constructed?

➤ Thanks. It is a good point. However, the major point of this paper is to illustrate a chemoenzymatic system that can turn CO₂, including gaseous and air-capture CO₂, as well as methanol into amino acid and pyruvate. The regeneration of DTT is

important, but it is beyond the scope of this story. We are working now on DTT regeneration, including the design of suitable catalyst and the optimization of electron flows, in follow-up studies.

2. Methanol needs to be incorporated in the title as they provides energy. Yes methanol can be derived from CO₂ at the expense of energy input (for instance H₂). But in the case of the authors, it would be misleading for the researchers if there is no methanol in the title – the reader would read the title as a novel process to turn CO₂ into organics without ATP and NADPH. However, the fact is that methanol has to be added to the process as a co-substrate, and methanol is the starting point for calculating the CO₂ fixation rate, therefore it cannot be neglected in the title.

➤ Thanks for the feedback.

We have changed the title to “Turn air-captured CO₂ with methanol into amino acid and pyruvate in an ATP/NAD(P)H-free chemoenzymatic system”.